# TCSurv:
# Time-based Clustering for Reliable Survival Analysis

## Abstract

Survival analysis is critical in healthcare for predicting time-to-event outcomes such as disease progression or patient survival. While deep learning excels at capturing meaningful representations from complex clinical data and has improved performance in deep survival models, it inherently struggles with reliability and robustness, challenges that are especially significant when deploying these models in real-world clinical practice. Out-of-distribution (OOD) detection, designed to identify or flag samples that deviate from the training distribution, has become a key method for evaluating AI reliability across fields. This capability is especially important in clinical applications, where noisy or heterogeneous patient data can lead to incorrect assessments; yet, OOD detection remains underexplored and challenging in deep survival analysis due to the need to handle both censored and observed samples, which are unique to this domain. In this study, we address this critical gap by introducing TCSurv, a novel time-based clustering approach for survival analysis that handles both observed and censored samples for robust OOD detection. TCSurv initializes cluster centers using in-distribution data, creating time-specific clusters that anchor model predictions for both observed and censored samples. Experiments in real-world clinical data, including Alzheimer's dementia progression, and benchmark medical imaging datasets demonstrate that TCSurv effectively distinguishes OOD samples without compromising survival performance compared to existing deep survival analysis frameworks. The full code is available at https://anonymous.4open.science/r/TCSurv-F585.

## 1 Introduction

Survival analysis is a critical statistical tool used to predict the time until the occurrence of a specific event, such as death, disease recurrence, or hardware failure (Klein & Goel, 2013; Clark et al., 2003; Allison, 2010). Notably, this is accomplished while accounting for *censored data*, which are elements in the dataset where the event time is unknown, representing an ambiguous regions in the data space. In healthcare, it offers a trajectory for disease progression until a critical point (i.e. disease onset or death). This is particularly relevant in cases like Alzheimer's dementia, for which there is currently no known cure and brain damage is irreversible, making early prediction through survival analysis essential for proactive clinical intervention.

In recent years, deep learning (DL) approaches have demonstrated exceptional capabilities in estimating complex survival functions from high-dimensional signals such as medical images (Wiegrebe et al., 2024). These advancements have enabled researchers to model disease progression from data sources like MR images (Thrasher et al., 2024), CT scans (Saeed et al., 2024), and whole-slide images (Xu et al., 2025). While DL-based survival analysis has achieved notable quantitative success, ensuring the reliability of these models remains a critical challenge (Zheng et al., 2023; Jafarzadeh et al., 2021). This concern is especially pressing in medical contexts, where, without reliability guarantees, clinicians may be unable to trust DL-based predictions.

One crucial aspect of ensuring reliability is the ability to detect out-of-distribution (OOD) inputs (Jafarzadeh et al., 2021), which are data samples that deviate significantly from the training distribution. OOD detection is essential because deep learning models can produce confident yet incorrect predictions when exposed to

unfamiliar, corrupted, or low quality inputs. For example, in survival analysis for Alzheimer's Disease (AD) progression, a model should be able to flag distorted or corrupted MR images for further clinical review to prevent misleading prognostic outputs. Without such safeguards, incorrect predictions could delay necessary interventions or lead to inappropriate treatment decisions. Therefore, developing robust OOD detection mechanisms is critical for ensuring reliable survival predictions and facilitating the safe translation of DL based models into real world healthcare practice.

In recent years, DL has seen the emergence of various frameworks aimed at identifying and rejecting OOD samples to enhance model reliability. However, most of these algorithms have been developed for relatively static tasks such as classification or detection (Yang et al., 2022; Cao et al., 2020; Shi & Lee, 2024), overlooking the unique challenges in survival analysis. In particular, survival learning objectives require specialized components which aggregate knowledge over multiple logits to account for data censorship. This results in deviations in learning dynamics compared to static domains, limiting the utility of current OOD detection approaches. For this reason, survival analysis demands novel methodologies to account for its time-to-event structure. To the best of our knowledge, no prior work has explicitly addressed OOD detection in the context of survival analysis.

Toward this, we propose TCSurv, a novel framework that leverages time-based clustering for OOD detection in survival analysis. TCSurv forms compact, time-specific clusters by guiding sample representations based on their event status: uncensored samples are pulled toward designated time-based anchor points, while censored samples are grouped with clusters beyond their censorship time, ensuring meaningful grouping even under censoring constraints. This approach enables the model to account for temporal dynamics and censoring patterns inherent in survival data. By organizing representations around meaningful time-based clusters, TCSurv enhances the survival model's ability to flag anomalous or unfamiliar inputs as OOD, thereby improving the real-world applicability of survival models.

We evaluate TCSurv across a diverse set of survival analysis datasets, including ADNI (MRI imaging for Alzheimer's disease), FLCHAIN (clinical lab data), GBSG (breast cancer clinical data), and METABRIC (gene expression and clinical features). Additionally, we test on six medical imaging datasets from the MedMNIST collection to further assess performance across modalities and organs. Our results demonstrate that TCSurv effectively distinguishes OOD samples while maintaining strong survival analysis performance, consistently outperforming existing deep survival analysis models. Through comprehensive experiments on diverse datasets, TCSurv shows remarkable robustness and generalizability across multiple medical imaging modalities and clinical data types. These extensive analyses not only validate the efficacy of our time-based clustering approach for OOD detection but also highlight its potential for reliable real-world deployment in survival prediction tasks.

Overall, our contributions are as follows: (1) For the first time, we conduct an OOD detection study within the context of deep survival models, addressing a significant gap in current survival analysis research. (2) We introduce TCSurv, a novel time-based clustering approach specifically designed to improve OOD detection in survival analysis. TCSurv achieves superior OOD detection performance compared to existing methods, without compromising predictive accuracy in survival tasks. (3) We conduct a comprehensive evaluation of our approach across a diverse set of survival analysis healthcare datasets, including a real-world Alzheimer's disease progression dataset, demonstrating the applicability and effectiveness of our proposed method for OOD detection and survival analysis in clinical settings.

## 2 Related Work

### 2.1 Survival analysis

Nonparametric deep survival analysis approaches have emerged as promising methods to model survival distributions. Unlike parametric approaches (Sheng & Henao, 2025; Lee et al., 2023), which make strong assumptions about the underlying distributions, or semi-parametric methods (Katzman et al., 2018), which follow a more restrictive proportional hazards assumption, nonparametric models seek to directly estimate the survival function without relying on such assumptions. This is achieved by decomposing the study window into $T$ discrete time intervals and estimating the probability that the event of interest will occur at each time

Figure 1: (Left) A deep model, $f_\theta(X)$, extracts features $z$ from an input $X$. (Center) A distance based loss is applied to the extracted $z$. Samples where the event is observed ($\delta = 1$) are pulled toward time cluster $t_k$ corresponding to the observation time, while the samples which are censored ($\delta = 0$) are pulled equally toward all time clusters after the time of censoring, reflecting the possibility of the event occurring at some (unknown) time in the future. (Right) OOD samples are embedded further away in the feature space.

$t \in \{0, 1, ..., T\}$. Under this formulation, (Lee & Whitmore, 2006) extend standard negative log-likelihood (NLL) to support right-censored data, whereas (Fotso, 2018b) instead approach it as a multi-task logistic regression (MLTR) problem. However, a study by (Kvamme & Ørnulf Borgan, 2019) claims that MLTR is practically equivalent to NLL, but with an additional reversed cumulative sum. Additionally, other works have noted that utilizing an additional ranking loss component, which seeks to learn the proper ordering of uncensored individuals, can be utilized as a powerful regularizer to further improve the discriminative performance of nonparametric survival models (Lee et al., 2018; Kamran & Wiens, 2021b). These strategies are being applied across various domains, including medical imaging (Nakagawa et al., 2020; Bello et al., 2019); however, the reliability of survival models has been lesser explored in recent literature, particularly in terms of OOD detection.

While deep survival analysis is never truly reliable due to its black box nature, recent efforts have worked toward establishing trustworthiness through uncertainty quantification, fairness analysis, and interpretability. For example, SurvUnc (Liu et al., 2025) introduces a meta-model framework which quantifies prediction uncertainty, providing a mechanism to identify lack of knowledge in a survival model. We note that uncertainty quantification is fundamentally different from OOD detection in that the former simply aims to evaluate model confidence, rather than explicitly identify distributional shifts in the data itself. Deep networks tend to suffer from overconfidence on distribution shifted data, limiting the utility of uncertainty quantification in such cases (Wei et al., 2022a). Other works (Zhang & Weiss, 2022; Zhang et al., 2023) have aimed to enhance reliability through fairness analysis to evaluate model bias toward disparate populations. Finally, approaches such as SurvSHAP (Krzyziński et al., 2023) and SurvLIME (Pachón-García et al., 2024) incorporate temporal mechanisms to traditional feature important analyses to extend them to the survival task. This improves the interpretability of deep survival models, thereby enhance overall reliability. Despite these efforts, there are alarmingly few works which consider reliability in terms of OOD detection.

## 2.2 Out-of-distribution detection

Distributional shifts due to data acquisition failures/corruptions, variations in pathology, and human error limit the reliability of deep learning models which are known to produce overconfident predictions on such elements. These deviations can be broadly categorized into *contextual-*, *semantic-* and *covariate-shift* classifications (Hong et al., 2024). Here, contextual-shift refers to variations in modality, semantic- (also called concept-) shift represents deviations in target class (e.g. novel pathologies), and covariate-shifts relate to differences in data acquisition, regional demographics, object orientation, or data corruption.

Given this, out-of-distribution (OOD) detection has become essential for assessing model reliability (Zheng et al., 2023; Jafarzadeh et al., 2021). Significant work has been done in this area, from post-processing techniques that analyze learned predictive distributions (e.g., MaxLogit (Zhang & Xiang, 2023)) to methods that normalize feature and logit spaces to distinguish in-distribution (ID) from OOD data (e.g., LogitNorm (Wei et al., 2022b)). Other approaches include open-set recognition and clustering-based methods (Liu et al., 2022; Sinhamahapatra et al., 2022). Additionally, benchmark studies have been established to understand and evaluate model performance across various datasets (Yang et al., 2022), including medical imaging applications (Cao et al., 2020) and geography based covariate-shift domains (Tosaki et al., 2025). However, these studies have primarily focused on classification and segmentation tasks, with alarmingly few exploring OOD detection in survival analysis. While some previous work have somewhat explored OOD detection within this domain (Loya et al., 2020; Liu et al., 2025), their scope is limited to evaluating OOD through uncertainty quantification, which is often insufficient due to the tendency of DL models to provide overconfident predictions regardless of the input distribution. In contrast, our study targets OOD detection specifically within the survival analysis framework. While it may appear that standard OOD detection techniques developed for classification tasks, such as Max Softmax Probability (MSP) (Hendrycks & Gimpel, 2017), could be readily applied, this is not the case in survival analysis. Survival models produce time-dependent risk estimates and censored outcomes rather than normalized class probabilities, rendering confidence-based detectors ill-defined and often misleading in this setting.

## 3 Methods

### 3.1 Preliminary I: Nonparametric Survival analysis

The objective of survival analysis is to model the time until a specific event occurs, such as disease progression or patient mortality, which is particularly relevant for medical prognosis. Formally, consider a dataset $\mathcal{D} = \{X_i, \delta_i, t_i \mid i \in \mathbb{N}\}$ of size $N$, where $X_i$ represents features extracted from medical data, $\delta_i \in \{1, 0\}$ is an indicator of whether the event of interest occurred within the observation window (1) or is right-censored (0), and $t_i$ denotes the time of event occurrence or censoring. Importantly, censorship implies that the event was not observed during the study period but may still occur later. The goal is to estimate the survival function $S(t) = P(T > t)$, which expresses the probability that the event has *not* occurred by time $t$. Under the nonparametric framework, survival models predict an individualized probability mass function (PMF) directly, which represents the probability of event occurrence at each time interval within the horizon. The *Cumulative Incidence Function* (CIF), which represents the probability of event occurrence at or before time $t$, is then computed as:

$$F(t|X) = \sum_{i=0}^{t} \phi(f_\theta(X))_i \tag{1}$$

where $\phi(\cdot)$ is the Softmax of the predicted PMF (Lee et al., 2018). Finally, the predicted survival function is simply the inverse of the CIF, expressed as: $S(t|X) = 1 - F(t|X)$ (Kvamme & Ørnulf Borgan, 2019). This provides a convenient method to obtain an individualized survival distribution directly without imposing restrictive distributional assumptions. We provide more details in Appendix A.

### 3.2 Preliminary II: OOD detection

OOD detection aims to identify inputs that are different than the training data distribution, a crucial capability in applications in the medical domain where unusual cases may indicate rare conditions, novel pathologies, or even noisy or faulty samples. Given a dataset $\mathcal{D} = \{X_i | i \in \mathbb{N}\}$ of size $N$, consisting of in-distribution samples, OOD detection seeks a score function $S_c(X)$ that quantifies how likely a new sample $X_{\text{new}}$ belongs to the training distribution (also referred as in-distribution (ID) data). A threshold $\phi$ is then defined, where $X$ is OOD if $S_c(X) > \phi$ and ID otherwise.

### 3.3 Time-based clustering

To enhance the robustness of survival analysis models, we present a novel approach that leverages time-based clustering to create well-defined feature spaces. We introduce a distance-based objective function specifically designed to learn representations that maintain clear separations between samples associated with different event times, ensuring that embeddings are closely clustered around time-specific anchors. This improves alignment with survival outcomes and naturally positions atypical samples farther from these clusters, resulting in a feature space that is inherently resilient to variations outside the training distribution. The overall framework is visualized in Figure 1

Toward this, we begin by discretizing survival times into $T$ equally spaced intervals, such that each time $t$ maps to a discrete time label $t \in \{0, 1, ..., T\}$. For each sample $\mathbf{X}^*$ with an observed event at time $t$, we aim to learn a feature representation $\mathbf{z}^* = f_\theta(\mathbf{X}^*)$, where $f_\theta(\cdot)$ is a deep learning model parameterized by $\theta$. The objective is to ensure that this embedding $\mathbf{z}^*$ is positioned close to a predefined anchor $c_t$. This setup encourages ID data to cluster around their respective time-based anchors, while OOD data is expected to lie farther from these anchors. To achieve this clustering effect, we propose the following objective function, which combines temporal separation and anchor proximity terms to guide embeddings toward their time-specific clusters:

$$\mathcal{L} = \mathcal{L}_T + \lambda \cdot \mathcal{L}_A \tag{2}$$

where $\mathcal{L}_T$ promotes separation between time clusters, and $\mathcal{L}_A$ ensures that embeddings remain close to their respective time anchors. However, survival data are progressive in nature. Depending on binning granularity, a subject (and subsequent embedding) could fall between two anchor points in the representation space. Thus, placing too much emphasis on $\mathcal{L}_A$ may limit TCSurv's ability to model this transitory behavior. As such, $\lambda$ is used as a hyperparameter that controls the relative influence of anchor proximity.

While similar distance-based objective functions have been used in the computer vision literature to learn class-specific representations for tasks like classification and detection (Miller et al., 2021), the survival analysis context presents unique challenges. Specifically, the presence of censored samples—instances where the exact event time is unknown—complicates the learning process, as these samples lack precise time labels yet still contribute valuable information. Consequently, our objective function must incorporate a specialized approach that distinguishes between censored and observed samples, ensuring that both types are effectively utilized in training.

As such, for *observed* data points, the calculation of $\mathcal{L}_T$ and $\mathcal{L}_A$ is straightforward, and is expressed as:

$$\mathcal{L}_\mathcal{T} = \log\left[1 + \sum_{j \neq t}^{T} \exp\left(d_t - d_j\right)\right], \tag{3}$$

$$\mathcal{L}_\mathcal{A} = ||f(X) - c_t||_2 \tag{4}$$

where, $d = \{||z - c_0||_2, ||z - c_1||_2, ..., ||z - c_n||_2\}$ is the distance vector which represents all of the distances between an embedding $z$ and every anchor, $n$ is the total number of clusters, and $t$ corresponds to the ground truth time anchor.

Whereas, for *censored* samples, we lack precise event times, making it challenging to assign them directly to a specific time anchor. However, useful survival analysis-based insights can still guide their embedding. For instance, if a sample $\mathbf{X}^*$ is censored at time $t = 3$, we know that $\mathbf{X}^*$ cannot correspond to the time anchors for $\{0, 1, 2, 3\}$ but may belong to any of the anchors in $\{4, 5, ..., T\}$. Additionally, a sample censored at the maximum observation time $t = T$ suggests that the event occurs beyond the observation window. To account for such cases, we introduce an extra time anchor at $t = T + 1$ to capture this uncertainty. To incorporate censored samples effectively, we define modified loss terms, $\mathcal{L}_{\mathcal{T}_c}$ and $\mathcal{L}_{\mathcal{A}_c}$ which extend the original objective in equations Eqns 3 and 4 to support the unique requirements of censored data, as follows:

$$\mathcal{L}_{\mathcal{T}_c} = \frac{\sum_{k=t+1}^{T+1} \log\left[1 + \sum_{j=0}^{t} \exp\left(d_k - d_j\right)\right]}{T - t}, \tag{5}$$

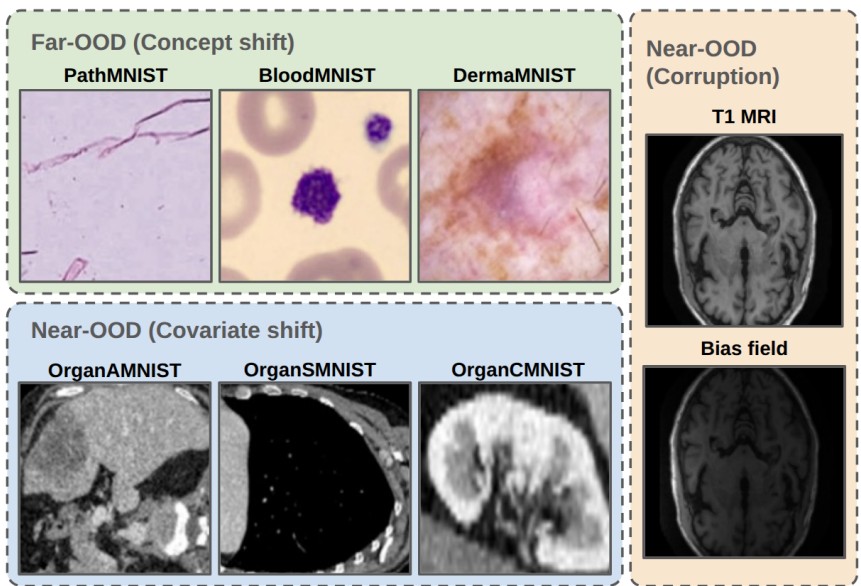

Figure 2: Examples of OOD scenarios considered in this study. {Path, Blood, Derma}MNIST represent a far-OOD, modality shift case, Organ{A, S, C}MNIST are near-OOD, covariate shift case, and MRI data correspond to OOD as it relates to data corruption.

$$\mathcal{L}_{\mathcal{A}_{\mathcal{C}}} = \frac{\sum_{k=t+1}^{T+1} ||f(x) - c_k||_2}{T - t}. \tag{6}$$

Finally, our combined time-based clustering objective ($\mathcal{L}$) for both censored and uncensored samples in the training data is defined as:

$$\mathcal{L} = \underbrace{\mathcal{L}_{\mathcal{T}} + \lambda_1 \mathcal{L}_{\mathcal{A}}}_{\text{Observed samples}} + \underbrace{\mathcal{L}_{\mathcal{T}_{\mathcal{C}}} + \lambda_2 \mathcal{L}_{\mathcal{A}_{\mathcal{C}}}}_{\text{Censored samples}} \tag{7}$$

Prior to training, the time anchors are initialized as the scaled one-hot vector $e_i, \forall i \in \{0, 1, ..., T+1\}$. The set of anchors is then defined as $C = \{c_0, ..., c_{T+1}\} = \{\alpha \cdot e_0, ..., \alpha \cdot e_{T+1}\}$, where $\alpha$ is a hyperparameter which defines the magnitude of the $e_i$. It is worth noting that $\lambda_1$ and $\lambda_2$ could be tuned individually, but we use a single $\lambda$ by setting $\lambda_1 = \lambda_2$ for our experiments.

### 3.4 TCSurv Evaluation

#### 3.4.1 Survival evaluation

TCSurv follows a more recent trend of directly predicting the survival function, in contrast to the proportional hazard assumption used in prior methods such as DeepSurv (Katzman et al., 2018). To achieve this, we begin by relocating the initial time anchors to the average locations of the correctly predicted elements. Specifically, for uncensored elements, a prediction is considered correct if it most closely aligns with the anchor corresponding to its ground truth event time. Since censored elements are more ambiguous, we define a prediction as correct if it is assigned to any time anchor *after* the time of censoring, allowing flexibility for model assignments of censored elements. We demonstrate that this action improves overall performance in Sec. 5.2.1. Since TCSurv is optimized to place embeddings closest to the anchors which most closely align with the event time, the PMF can be readily computed by taking the soft*min* of the distance vector. Softmin will yield higher probabilities for anchors which are nearest to the embedding and lower values for the ones which are further away. From here, the survival function can be obtained via the relationships discussed in Sec. 3.1:

| Models | C-td (Survival performance) | AUROC (OOD performance) | |
| | | ADNI-T2 | Bias |
| --- | --- | --- | --- |
| NLL | $0.7786 \pm .047$ | $0.3361 \pm .050$ | $0.5361 \pm .028$ |
| N-MLTR | $0.8128 \pm .015$ | $0.4560 \pm .069$ | $0.4704 \pm .051$ |
| RPS | $0.8668 \pm .025$ | $0.4041 \pm .017$ | $0.4992 \pm .037$ |
| RPS+Rank | $0.8576 \pm .023$ | $0.3475 \pm .036$ | $0.4391 \pm .029$ |
| DeepHit | $0.7770 \pm .026$ | $0.4819 \pm .067$ | $0.4105 \pm .044$ |
| SurvRNC | $0.6528 \pm .079$ | $0.4806 \pm .058$ | $0.4118 \pm .036$ |
| TE-SSL | $0.6103 \pm .018$ | $0.4690 \pm .047$ | $0.3789 \pm .032$ |
| TCSurv (Ours) | $\mathbf{0.8798} \pm \mathbf{.017}$ | $\mathbf{0.7165} \pm \mathbf{.080}$ | $\mathbf{0.5765} \pm \mathbf{.014}$ |

Table 1: (Left) Survival performance. (Right) OOD performance for ADNI data.

$$\hat{S}(t|X) = 1 - \sum_{i=0}^{t} \sigma(d)_i \tag{8}$$

where $\sigma(\cdot)$ is the soft$min$ function applied to the distance vector. We then quantify our model's discriminative survival performance based on the time-dependent concordance index (C-td) (Antolini et al., 2005) (detailed in Appendix A.1), enabling a fair comparison of our method against existing approaches.

### 3.4.2 OOD evaluation

For OOD evaluation, the central idea is that the further away any new input $X_{\text{new}}$ is from the time-based anchors, the more likely it is to be an OOD sample. A straightforward way to achieve this would be to simply use the distance $d$ (which represents distance to all the anchors) as the score function. However, prior works in the classification domain (Miller et al., 2021) have considered the combination of $d$ and softmin($\cdot$) function to achieve a more robust score function. Hence, we also consider a similar strategy for designing the score function for evaluating OOD detection with TCSurv as:

$$S_c(X) = \min\big(d \cdot (1 - \sigma(d))\big). \tag{9}$$

This score function will have minimum value when the inputs have both a low distance and high softmin score. We note that, since baseline methods do not have a notion of time anchors. However, we can utilize the predicted PMFs from these approaches to compute the "learned anchor" positions in a similar fashion to the method described in Sec. 3.4.1. Specifically, for these approaches, we consider a prediction to be correct if the probability is maximized at the event time for observed elements, or any time after censorship for censored elements. This allows us to fit our OOD detector to other survival methods for fairer comparison.

## 4 Data

We validate the performance of TCSurv across a range of datasets which include real-world 3D MRI data, real-world non-imaging data, and a synthetic survival analysis dataset using benchmark medical imaging data. Together, these datasets provide insights into TCSurv's performance across modality shift, data corruption, and cross-dataset shift.

### 4.1 Alzheimer's Disease progression MRI dataset

We utilize MRI data from the Alzheimer's Disease Neuroimaging Initiative (ADNI) (Petersen et al., 2010) for our analysis on real world medical imaging data. ADNI provides follow up information on patients' progression towards Alzheimer's dementia. This dataset includes 1,978 T1-weighted 3D MRI scans used as our ID data, and 919 T2-weighted MRIs treated as OOD. Since survival models are usually trained and evaluated on high-quality MRIs, artifacts from equipment errors naturally represent OOD data. To this end, we also generate additional OOD samples which simulate intensity inhomogeneity, known as bias fields (shown in Fig. 2). These variations create a blurring effect on the MRI, which inhibits feature detection, weakening model performance (Jindal et al., 2022). We generated these artifacts using the RandomBias augmentation pipeline from the Torchio library (Pérez-García et al., 2021).

|  | C-td | ID:FLCHAIN | |
| Model | FLCHAIN | GBSG | METABRIC |
|---|---|---|---|
| NLL | **0.8328 ± .004** | 0.4371 ± .032 | 0.4091 ± .030 |
| N-MTLR | 0.8321 ± .006 | 0.4003 ± .020 | 0.4189 ± .036 |
| RPS | 0.8126 ± .005 | 0.4810 ± .021 | 0.4565 ± .026 |
| RPS+Rank | 0.8197 ± .007 | 0.4632 ± .063 | 0.4514 ± .048 |
| DeepHit | 0.8389 ± .002 | 0.4488 ± .054 | 0.4610 ± .038 |
| SurvRNC | 0.7831 ± .009 | 0.4892 ± .003 | 0.4596 ± .008 |
| TE-SSL | 0.7816 ± .009 | 0.4772 ± .006 | 0.4453 ± .018 |
| TCSurv | 0.8229 ± .002 | **0.7010 ± .041** | **0.7000 ± .021** |

Table 2: (Left) Survival performance (Right) OOD performance for low-dimensional benchmark non-imaging data.

| Model | C-td (Survival Performance) | AUROC (OOD Performance) |
|---|---|---|
| NLL | **0.8737 ± 0.0011** | 0.4440 ± 0.0473 |
| N-MTLR | 0.8721 ± 0.0004 | 0.3961 ± 0.1346 |
| RPS | 0.8696 ± 0.0005 | 0.5870 ± 0.0598 |
| RPS + Ranking | 0.8694 ± 0.0010 | 0.5253 ± 0.0967 |
| DeepHit | 0.8720 ± 0.0003 | 0.3412 ± 0.0755 |
| SurvRNC | 0.8605 ± 0.0002 | 0.6078 ± 0.0840 |
| TESSL | 0.8508 ± 0.0135 | 0.5201 ± 0.0297 |
| TCSurv | 0.8684 ± 0.0025 | **0.7226 ± 0.0772** |

Table 3: (Left) Survival performance (Right) OOD performance for high-dimensional NACC data.

## 4.2 Non-imaging data

To reinforce our results, we include additional non-imaging data from *three* standardized survival datasets: 1) Assay of Serum Free Light Chain (**FLCHAIN**) (Dispenzieri et al., 2012), 2) German Breast Cancer Study (**GBSG**) (Schumacher et al., 1994), and 3) the Molecular Taxonomy of Breast Cancer International Consortium (**METABRIC**) (Curtis et al., 2012). These continuous time datasets are then discretized with the pipeline laid out in (Kvamme & Ørnulf Borgan, 2019). For our experiments, we train on FLCHAIN and treat GBSG and METABRIC as OOD, truncating and padding features as necessary to align with model dimensions. Since the aforementioned benchmark datasets are relatively low in complexity (due to their limited number of features) and somewhat low levels of censorship, we include additional experiments using real world data from the National Alzheimer's Coordinating Center (NACC) (Kukull, 2025) to further demonstrate the effectiveness of TCSurv on complex real-world data. As it would be infeasible to utilize the benchmark nonimaging datasets as OOD scenarios without severe padding, we randomly permute the columns of NACC to simulate an OOD dataset. We provide further details on these datasets in Appendix B.

## 4.3 Medical imaging benchmark dataset

We also include medical imaging datasets from the MedMNIST (Yang et al., 2023) collection. In specific, we utilize six different datasets, which are divided into two groups. In the first group, we analyze the far-OOD performance on three different modalities: Colon Pathology (**PathMNIST**), Blood Cell Microscope (**BloodMNIST**), and Dermatoscope (**DermaMNIST**) imaging. The second group analyzes near-OOD detection performance when exposed to different 2D views of the same 3D abdominal CT images, namely the Axial (**OrganAMNIST**), Saggital (**OrganSMNIST**), and Coronal (**OrganCMNIST**) views. We provide some examples in Fig. 2 and more details on their statistics in Appendix B.

Synthesizing survival signals: Since benchmark medical datasets lack survival information, we synthesize survival signals by assigning event times and censoring indicators to each class. This overcomes the scarcity,

|  | C-td | | ID:SurvPath | | ID:SurvBlood | |
|  | SurvPath | SurvBlood | SurvBlood | SurvDerma | SurvPath | SurvDerma |
| --- | --- | --- | --- | --- | --- | --- |
| NLL | $0.9773 \pm .004$ | $\mathbf{0.9954 \pm .001}$ | $0.4985 \pm .004$ | $0.5004 \pm .007$ | $0.5452 \pm .011$ | $0.5197 \pm .007$ |
| N-MTLR | $0.9881 \pm .002$ | $0.9958 \pm .001$ | $0.5076 \pm .021$ | $0.5322 \pm .025$ | $0.5731 \pm .018$ | $0.5174 \pm .016$ |
| RPS | $0.9733 \pm .002$ | $0.9936 \pm .001$ | $0.5160 \pm .004$ | $0.4907 \pm .004$ | $0.5459 \pm .009$ | $0.5227 \pm .008$ |
| RPS+Rank | $0.9740 \pm .002$ | $0.9927 \pm .001$ | $0.5125 \pm .011$ | $0.4929 \pm .008$ | $0.5716 \pm .015$ | $0.5256 \pm .016$ |
| DeepHit | $\mathbf{0.9861 \pm .002}$ | $0.9951 \pm .001$ | $0.5141 \pm .023$ | $0.5113 \pm .011$ | $0.5717 \pm .008$ | $0.5172 \pm .011$ |
| SurvRNC | $0.9722 \pm .003$ | $0.9899 \pm .001$ | $0.6131 \pm .035$ | $0.5108 \pm .044$ | $0.6123 \pm .035$ | $0.6335 \pm .141$ |
| TE-SSL | $0.9660 \pm .001$ | $0.9878 \pm .001$ | $0.5265 \pm .034$ | $0.4644 \pm .020$ | $0.6479 \pm .048$ | $0.5209 \pm .012$ |
| TCSurv (Ours) | $0.9813 \pm .001$ | $0.9809 \pm .002$ | $\mathbf{0.7989 \pm .042}$ | $\mathbf{0.7649 \pm .016}$ | $\mathbf{0.8933 \pm .020}$ | $\mathbf{0.9403 \pm .016}$ |
|  | C-td | | ID:SurvOrganA | | ID:SurvOrganS | |
|  | SurvOrganA | SurvOrganS | SurvOrganS | SurvOrganC | SurvOrganC | SurvOrganC |
| NLL | $\mathbf{0.9927 \pm .001}$ | $0.9671 \pm .004$ | $0.5273 \pm .005$ | $0.5177 \pm .004$ | $0.5272 \pm .000$ | $0.5129 \pm .001$ |
| N-MTLR | $0.9921 \pm .001$ | $0.9678 \pm .004$ | $0.5617 \pm .018$ | $0.5406 \pm .011$ | $0.5358 \pm .012$ | $0.5175 \pm .004$ |
| RPS | $0.9892 \pm .001$ | $0.9682 \pm .002$ | $0.5515 \pm .006$ | $0.5463 \pm .005$ | $0.5280 \pm .006$ | $0.5197 \pm .005$ |
| RPS+Rank | $0.9920 \pm .002$ | $0.9708 \pm .003$ | $0.5425 \pm .017$ | $0.5359 \pm .012$ | $0.5267 \pm .003$ | $0.5153 \pm .000$ |
| DeepHit | $0.9918 \pm .002$ | $\mathbf{0.9724 \pm .002}$ | $0.5276 \pm .005$ | $0.5214 \pm .004$ | $0.5364 \pm .014$ | $0.5137 \pm .005$ |
| SurvRNC | $0.9853 \pm .003$ | $0.9627 \pm .004$ | $0.6138 \pm .045$ | $0.5737 \pm .031$ | $0.5911 \pm .016$ | $0.5276 \pm .004$ |
| TE-SSL | $0.9878 \pm .002$ | $0.9582 \pm .018$ | $0.5525 \pm .011$ | $0.5316 \pm .003$ | $0.5853 \pm .036$ | $0.5229 \pm .012$ |
| TCSurv (Ours) | $0.9809 \pm .002$ | $0.9496 \pm .002$ | $\mathbf{0.7453 \pm .005}$ | $\mathbf{0.7032 \pm .011}$ | $\mathbf{0.6098 \pm .023}$ | $\mathbf{0.5572 \pm .011}$ |

Table 4: (Left) Survival performance (Right) OOD performance for medical imaging benchmark datasets

noise, and heavy censoring of real-world survival data and provides a controlled environment with adjustable censorship and known ground truth. For each $(X_i, y_i)$ image-label pair, the ground truth event time $t_i^*$ is initially set to the class label $y_i$. We then uniformly censor a portion of the data by setting $\delta_i = 0$. However, since, by definition of censorship, it is known that an element cannot be censored at its ground truth event time, we shift the time of censorship to a uniformly random $t \in \{0, 1, ..., t_i^* - 1\}$. Conversely, elements with $t_i^* = 0$ are always observed and are made to be uncensored. Finally, we completely censor all elements where $t_i^* = \max(T)$ to allow the model to provide elements which should be assigned to the "future" time cluster. We prefix these datasets with "Surv-" and drop the "MNIST" (e.g., SurvPath) to distinguish them from standard MedMNIST datasets. We also use "Surv*" as shorthand for the entire set of synthetic survival datasets for simplicity.

## 5 Experiments

This section presents our experimental results and ablation analyses. We begin with a description the experimental settings and baselines and then report survival prediction, OOD detection results, and ablation studies to validate the framework.

### 5.1 Experimental setup

We evaluate our method against Negative Log-Likelihood (**NLL**) (Kvamme & Ørnulf Borgan, 2019), Neural Multi-Task Logistic Regression (**N-MTLR**) (Fotso, 2018a), and Ranked Probability Scoring (**RPS**) (Kamran & Wiens, 2021a) models. Additionally, some works have explored the use of an additional ranking component, which seeks to learn the proper ordering of uncensored individuals. As such, we include **DeepHit** ($\mathcal{L}_{NLL} + \mathcal{L}_{Ranking}$) (Lee et al., 2018) and **RPS+Rank** ($\mathcal{L}_{RPS} + \mathcal{L}_{Ranking}$) (Kamran & Wiens, 2021a) in our baseline suite. Finally, since TCSurv can be viewed as a representation learning method, we include two recent approaches: **SurvRNC** ($\mathcal{L}_{DeepHit} + \gamma \mathcal{L}_{SurvRNC}$) (Saeed et al., 2024) and **TE-SSL** ($\mathcal{L}_{DeepHit} + \gamma \mathcal{L}_{TESSL}$) (Thrasher et al., 2024), to ensure that our improved OOD detection is not merely due to better feature learning, but rather to learning a more optimal feature space.

All models were optimized using the Adam algorithm (Kingma, 2014) with a learning rate $\Phi$. We use a 3D CNN (Liu et al., 2020) as the backbone model for ADNI, ResNet18 for the Surv* scenario, and a feed forward neural network (Kvamme & Ørnulf Borgan, 2019) for the nonimaging datasets. ADNI were trained with

| | | SurvPathMNIST | | | SurvOrganAMNIST | | |
|---|---|---|---|---|---|---|---|
| | | C-td | AUROC (SurvBloodMNIST) | AUROC (SurvDermaMNIST) | C-td | AUROC (SurvOrganSMNIST) | AUROC (SurvOrganCMNIST) |
| $\alpha$ | 0.1 | 0.9725 | 0.7292 | **0.8371** | 0.9583 | 0.6743 | 0.6510 |
| | 1 | **0.9836** | **0.8434** | 0.7914 | **0.9794** | 0.7495 | 0.7132 |
| | 5 | 0.9671 | 0.7086 | 0.4976 | 0.9548 | **0.7499** | **0.7163** |
| | 10 | 0.9295 | 0.5430 | 0.3329 | 0.9021 | 0.7342 | 0.6919 |
| $\lambda$ | 0 | 0.8557 | 0.5200 | 0.3724 | 0.9299 | 0.7403 | 0.6912 |
| | .25 | **0.9837** | 0.8371 | 0.7834 | **0.9860** | 0.7838 | 0.7356 |
| | .5 | 0.9836 | 0.7832 | 0.8082 | 0.9844 | 0.7659 | 0.7258 |
| | .75 | **0.9837** | **0.8751** | **0.8505** | 0.9828 | 0.7536 | 0.7157 |
| | 1 | 0.9832 | 0.8112 | 0.8433 | 0.9809 | **0.7909** | **0.7498** |

Table 5: Ablation of $\alpha$ and $\lambda$ for Surv{Path, OrganA}.

$\alpha = 1$, $\lambda = 0.5$, $\Phi = 1 \times 10^{-4}$, Surv* with $\alpha, \lambda = 1$, $\Phi = 1 \times 10^{-4}$, and FLCHAIN with $\alpha, \lambda = 1$, $\Phi = 1^{-3}$. All experiments were conducted using NVIDIA A30, A40, and RTX 4500 Ada Generation GPUs. Results are reported as the mean and standard deviation across three random seeds. We provide more details on each baseline model in Appendix C.

## 5.2 Results

We present the primary results for the ADNI, non-imaging, and Surv* datasets in Tables 1, 3, and 4, respectively. In each table, we show the OOD results (AUROC) on the right and the corresponding survival results (C-td) on held-out in-distribution data on the left.

First, as shown across nearly all experiments, the OOD detection performance of TCSurv is significantly higher, achieving up to a 49% increase in the ADNI experiments, 43% with non-imaging data, and 48% with the Surv* suite compared to top performing baselines. This improvement in OOD detection can be attributed to TCSurv's explicit design that encourages in-distribution (ID) training data to form tight clusters around time-dependent anchor points. To empirically observe this behavior, we plot the distance to the nearest time-based center for our method in Figure 3 and compare it to the baselines. Here, we clearly observe that baseline methods fail to distinguish between ID and OOD data. In some cases (e.g., NLL), the logit space even places ID data farther from the center than OOD data. In contrast, TCSurv effectively achieves tighter clusters for ID data, while OOD data remain more dispersed. We additionally provide distance plots for the representative non-imaging experiments in the Appendix D.

Second, in terms of survival performance, TCSurv demonstrates results that are comparable and competitive with existing baselines. With the exception of GBSG and SurvOrganS, the survival outcomes remain closely aligned with those of the strongest baseline methods, with only marginal differences. Importantly, as discussed earlier, TCSurv consistently outperforms baselines in OOD detection across all datasets. This is particularly critical in real-world applications, where the reliability and robustness of a model are just as important as its accuracy.

### 5.2.1 Ablation analysis

$\alpha$ & $\lambda$ analysis: We examine the effect of $\alpha$, which controls the magnitude of the class anchor, and $\lambda$, which balances the contributions of $\mathcal{L}_{\mathcal{T}}$ and $\mathcal{L}_{\mathcal{A}}$, on both survival performance and OOD detection. The results, presented in Table 5 (top), indicate that the model generally prefers smaller $\alpha$ values, suggesting that a compact feature space is well-suited for both survival analysis and OOD detection.

In Table 5 (bottom), we analyze the effect of $\lambda$ to understand the role of anchor loss. As seen for both datasets and in both survival analysis and OOD performance, when $\lambda = 0$ (corresponding to zero anchor loss), the performance is poor, demonstrating the importance of $\mathcal{L}_A$. Meanwhile, as $\lambda$ increases, TCSurv tends to suffer in terms of survival capability. Despite the general improvement in OOD detection performance due

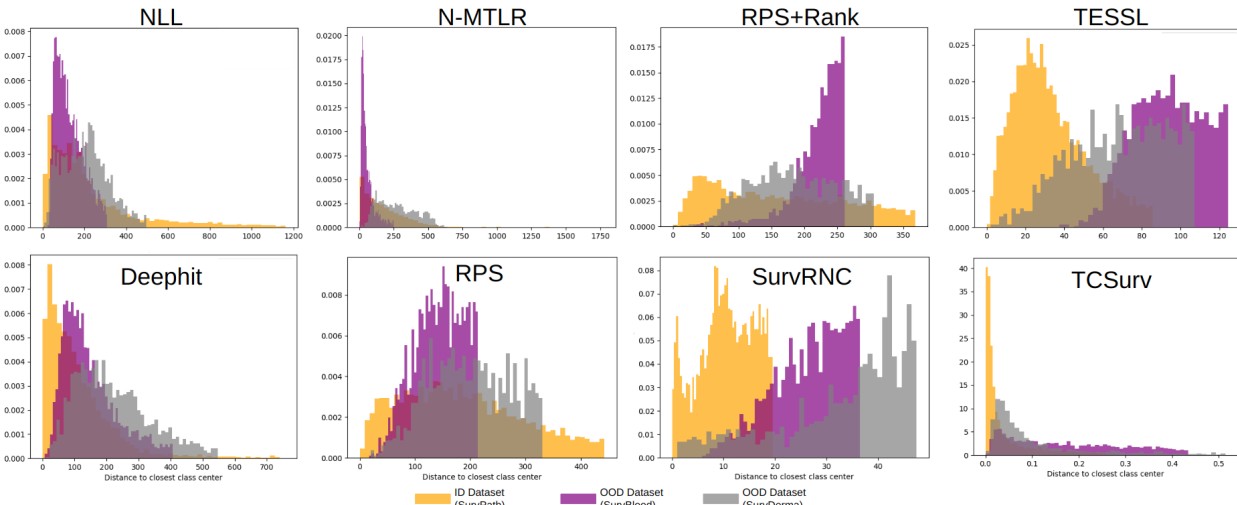

Figure 3: TCSurv allows the ID dataset to cluster more tightly around the time-based centers, while the distances of OOD samples to the nearest time-based centers are more spread out compared to the baselines.

to larger values of $\lambda$, this suggests that placing too much emphasis on cluster tightness around the anchors hinders the model's ability to properly represent progressive data.

Censored samples in determining the anchor position: TCSurv repositions the time anchors based on correctly predicted samples. As discussed in Sec 3.4.1, we consider both observed and censored elements for relocation, rather than updating centers solely based on observed ones. As shown in Fig. 4, while including censored samples does not significantly impact survival performance on benchmark datasets, it proves crucial for all OOD evaluations, where censored data plays an important role in determining accurate anchor positions. This is particularly apparent in the evaluation of ADNI-T2 data, where the model is entirely unable to predict OOD samples when anchors are relocated based solely on observed data.

Censorship ratio: As discussed in Sec. 4.3, synthetic data allows us to conduct a more thorough analysis of our model's behavior as the level of censorship varies. As such, Figure 5 depicts TCSurv's performance across varying levels of censorship. From the four plots on the right, it can be seen that our method consistently achieves superior AUROC on the OOD detection task when compared to the baseline models. It can also be seen from the two leftmost plots that survival performance from TCSurv maintains steady and competitive performance. While it is slightly below some of the baseline methods, we argue that TCSurv still performs strongly overall on the survival task, and that the massively improved OOD performance justifies a slight loss in the survival task since reliability is equally crucial in real-world applications. But we also acknowledge this as a limitation of our method as Eqs. 5 and 6 handle censored data in a more restrictive manner than the unbounded approaches in the baseline models. Finding a better balance between OOD detection performance and a less restrictive mechanism for censored data could be a valuable area for future work.

## 6    Discussion

In this work, we address the problem of OOD detection head-on in survival analysis for the first time by introducing TCSurv, a novel time-based clustering framework. TCSurv organizes representations of both observed and censored samples into distinct, compact time-based clusters, enhancing the model's ability to distinguish ID images from those in unseen distributions, thus improving OOD detection in survival analysis. The approach incorporates an objective function that promotes separation between time clusters while ensuring that learned embeddings remain close to their respective time anchors. Through extensive evaluations across a real-world MRI dataset, three clinical lab feature datasets, and six medical imaging benchmark datasets, TCSurv consistently improves OOD detection without compromising survival performance. Further, our experiments demonstrate that TCSurv consistently creates compact clusters around the designated

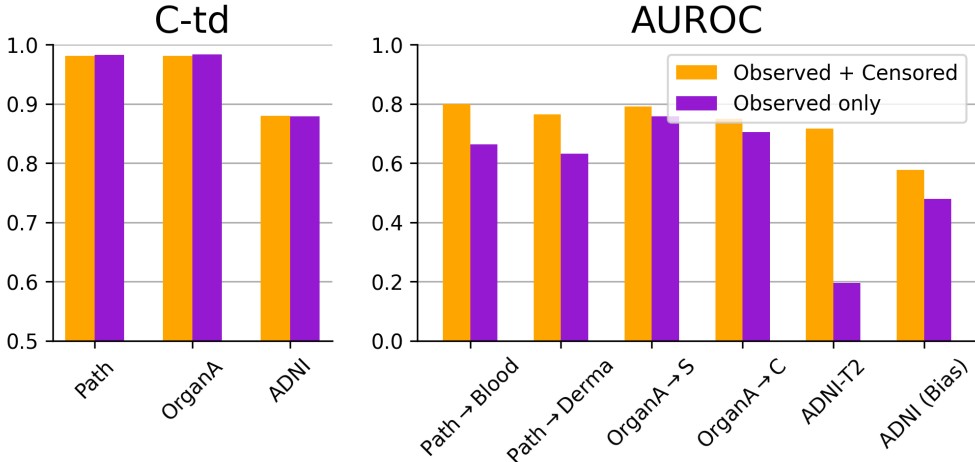

Figure 4: Analysis of the effect of including censored samples when updating anchor locations for both survival performance (left) and OOD detection (right).

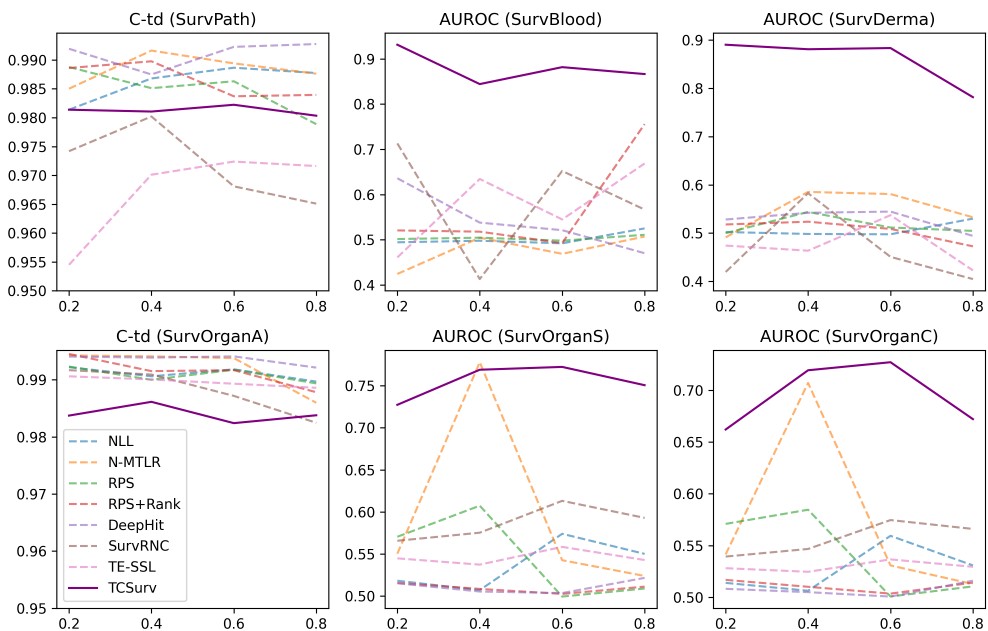

Figure 5: Analysis of model performance with varying levels of censorship. (Top) Trained on SurvPath as ID data and Surv{Blood, Derma} as OOD data. (Bottom) Trained on SurvOrganA as ID data and SurvOrgan{S, C} as OOD data.

time anchors and is robust the high degrees of censorship. Among the OOD detection tasks studied, the ability of our method to identify artifact-filled MRIs as OOD is particularly significant, as such artifacts, caused by equipment interference, can severely degrade image quality and diagnostic accuracy (Noda et al., 2022).

However, this study has certain limitations that point to important directions for future research. While prior work has explored OOD detection in survival analysis using epistemic uncertainty quantification (Loya et al., 2020; Liu et al., 2025), this work does not focus on uncertainty-based approaches. A key reason is that uncertainty estimates alone do not reliably capture distributional shifts in unseen data, as modern

deep neural networks are often poorly calibrated and tend to exhibit overconfident predictions even under significant distributional mismatch (Wang et al., 2021). In addition, our experimental formulation simplifies the clinical progression modeled in the ADNI dataset, which contains labels for Cognitively Normal (CN), Mild Cognitive Impairment (MCI), and Alzheimer's Dementia (AD) samples. To frame it as a survival task, we merged CN and MCI into a single category, predicting AD directly. Extending TCSurv to handle longitudinal survival data, thereby modeling transitions from CN to MCI to AD, could be a valuable future direction with significant translational implications. Since longitudinal methods follow a single subject over time, ensuring reliability across multiple visits is another important challenge to consider. Additionally, we note that the current anchor initialization method yields equidistant time centers, which may imply that all event times are equally spaced (i.e. $t = 1$ is as close to $t = 2$ as it is $t = 10$). While this formulation is effective, organizing anchors such that centroids are closer to temporally nearby anchors than distance ones could offer further improvement to the survival modeling process. We also note that since TCSurv is designed as a nonparametric approach, future work should explore OOD detection for parametric and semi-parametric methods. Lastly, our analysis using the Surv* datasets could be expanded to develop a standardized benchmark for survival-based OOD detection. This would likely attract broad interest and help drive progress in addressing this critical reliability challenge.

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

# Appendix

## A  Survival analysis

Survival analysis is a statistical tool used to model the time it takes until a specific event occurs. Survival data are collected in follow-up studies, where a subject is monitored until the event of interest has occurred, at which point the time between the initial visit and event occurrence is recorded as the survival outcome. Importantly, not all survival outcomes can be recorded due to the study ending prior to event observation or a subject dropping out of the study for some other reason. These instances are called *right-censored* samples, and are a crucial aspect to survival analysis because it is unknown if/when the event will occur in the future. In these cases, we instead record the last visit as the time of censorship ($\delta = 0$) since it is at least known that this subject has not experienced an event by this time.

Formally, consider subject data $X$. The ground truth event time can be sampled from a distribution $t^* \in \mathcal{T}$. Similarly, the time of censorship can be sampled from $c \in \mathcal{C}$. We choose the subject's event indicator to be $\delta = 1$ if $t^* \leq c$ and 0 otherwise. Then, the corresponding event time is $t = \min(t^*, c)$. Note that it is typically assumed that $\mathcal{T} \perp\!\!\!\perp \mathcal{C}|X$ for right-censored data.

### A.1  Survival evaluation

The Concordance Index (CI) evaluates the model's ability to correctly rank the event times of two samples, given the information. However, CI is only computed at the initial time of observation and does not reflect the change in risk over time (Lee et al., 2018). Instead, the time-dependent concordance index (C-td) evaluates the average concordance across *each time interval* to provide a more holistic view of the risk across the study. To compute C-td, we first must define the Cumulative Incidence Function (CIF), which expresses the probability that the event will occur on or before some time $t$. This is essentially the cumulative sum of the $S(t|X)$, and is defined as:

$$F(t|X) = \sum_{i=0}^{t} \sigma(f_\theta(X))_i \tag{10}$$

where $\sigma(\cdot)$ is the Softmax (or softmin for TCSurv) function. We then define C-td as:

$$Ctd = \frac{A_{i,j} * 1(F(t_i|X_i) > F(t_i|X_j))}{\sum_{i \neq j} A_{i,j}} \tag{11}$$

Here, $A_{i,j}$ is a function which determines whether a pair of elements $i, j$ are comparable, defined as $A_{i,j} = 1(\delta_i = 1, t_i < t_j)$. Higher is better for C-td, where 1 corresponds to perfect ranking, 0.5 is complete randomness, and 0 is inversely perfect.

## B  Detailed dataset descriptions

### B.1  ADNI

The ANDI dataset includes a cohort of 493 unique patients from the Alzheimer's Disease Neuroimaging Initiative (ADNI) (Petersen et al., 2010). Each participant has one or more visits that include 3D T1-weighted MRI scans, resulting in a total of 1,978 data points. A subset of patient visits additionally include a T2-weighted MRI, which we utilize as OOD data during evaluation. At each visit, patients are classified as cognitively normal (CN), having mild cognitive impairment (MCI), or diagnosed with Alzheimer's dementia (AD). We designate as converters those individuals who were initially CN or MCI but developed AD over the course of the study. Additionally, each visit records the number of months since the baseline observation, serving as a time-to-event indicator, with a 68.7% censor rate. The data underwent preprocessing following

| Dataset | Modality (#classes) | # Train / Val / Test |
|---|---|---|
| Path | Colon Pathology (9) | 89,996/10,004/7,180 |
| Blood | Blood Cell Microscope (8) | 11,959/1,712/3,421 |
| Derma | Dermatoscope (7) | - / - /2,005 |
| OrganA | Abdominal CT - Axial (11) | 34,561/6,491/17,778 |
| OrganS | Abdominal CT - Sagittal (11) | 13,932 / 2,452 / 8,827 |
| OrganC | Abdominal CT - Coronal (11) | - / - / 8,216 |

Table A1: Overview of medical imaging benchmark datasets (Yang et al., 2023).

the pipeline outlined in (Liu et al., 2020) and were split by unique participants to prevent data leakage. For patients with multiple visits, each visit is treated as an independent data point.

## B.2 Nonimaging data

All nonimaging datasets included in this study are continuous time survival datasets which were binned into 10 time windows using LabTransDiscreteTime from the Pycox (Kvamme & Ørnulf Borgan, 2019) library. For OOD evaluation, we truncated/padded the features to align every dataset with the model, as needed.

### B.2.1 FLCHAIN

This dataset contains 7874 individuals with 7 features over a time horizon of 5166 days and studies the relationship between serum free light chain (FLC) and mortality. FLCHAIN is approximately 69.9% right-censored.

### B.2.2 GBSG

GBSG contains follow up information for 473 patients with a 56 month time horizon and 7 features. The event of interest for GBSG is death due to breast cancer. Finally, GBSG has is 43.2% censored.

### B.2.3 METABRIC

METABRIC also contains follow up information for breast cancer mortality. It contains 9 gene expression features with a time horizon of 355 months and a censor rate of 42.1%.

### B.2.4 NACC

The National Alzheimer's Coordinating Center (NACC) contains over 200,000 records of follow-up from 55,000 unique patients, each with approximately 1,000 unique features. We preprocess this dataset by first assigning survival labels based on a subject's AD progression. Much like ADNI, we define *converters* as any patient who transitions from a non-AD state to an AD one and record the event time (or time of censorship) as the time between their first visit and their subsequent diagnosis. This resulted in a final censorship rate of 76.8%. Since NACC contains many redundant and uninformative features, we manually selected representative features based on their relevance to AD diagnosis and alignment to previous AD studies using NACC data (Sharma et al., 2021). Finally, the data underwent a standard preprocessing pipeline which included truncation of survival data, imputation, normalization, and one-hot-encoding.

## B.3 Synthetic data

Table A1 provides more details on MedMNIST datasets that were used to generate the synthetic data. Additionally, Figure A1 provides a visualization of our synthetic data generation approach.

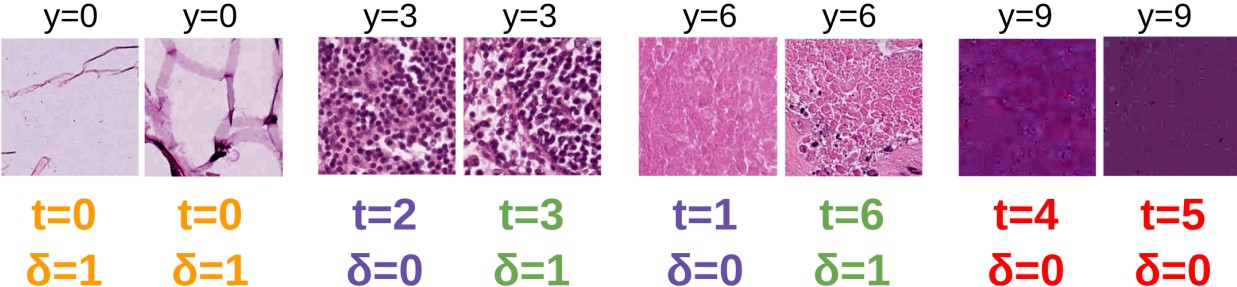

- **Observed elements are assigned event time $t_i = y_i$**
- **Elements with ground truth event time at $t* = 0$ are *always observed***
- **Censored elements are assigned a random event time $t < t*$**
- **All elements with ground truth event time $t* = T$ are *always censored***

Figure A1: Schematic outlining synthetic data generation, where $y$ is the class label, $t$ is the time of event/censoring, $t^*$ is the *ground truth* event time, and $\delta$ is the event indicator.

| | ADNI | FLCHAIN | Surv* |
|---|---|---|---|
| Optimizer | Adam | Adam | Adam |
| Learning rate | $1 \times 10^{-4}$ | $1 \times 10^{-3}$ | $1 \times 10^{-4}$ |
| Batch Size | 32 | 32 | 128 |
| $\alpha$ | 1 | 1 | 1 |
| $\lambda$ | 0.5 | 1 | 1 |
| $\gamma_1$ | 0.5 | 0.25 | 0.5 |
| $\gamma_2$ | 0.2 | 0.25 | 0.5 |

Table A2: Complete hyperparameter configuration for experiments.

## C  Futher implementation details

Table A2 outlines all of the hyperparameter configurations used in our experiments. We now provide more details on the baseline methods, hyperparameters, and model architectures.

### C.1  Baseline methods

We now provide more details on the baseline methods used for comparison.

### C.2  Standard survival methods

### C.2.1  Negative Log Likelihood (NLL)

This is an adaptation of log-likelihood proposed by (Lee & Whitmore, 2006) for right-censored survival data, defined as:

$$\mathcal{L}_{NLL} = -\sum_{i=1}^{B} \delta_i \log \hat{y}_i + (1 - \delta_i) \log \sum_{j=t_i+1}^{T} \hat{y}_j \tag{12}$$

where $B$ is the batch size and $\hat{y}_i = \text{softmax}(f_\theta(X_i))$. Specifically, for observed elements, $\mathcal{L}_{NLL}$ seeks to maximize the predicted probability at the observed event time. Then, for censored elements, it instead maximizes the sum of all probabilities *after* the time of censorship.

### C.2.2 Neural Multi-task Logistic Regression (N-MTLR)

According to (Kvamme & Ørnulf Borgan, 2019), N-MTLR is practically the equivalent to NLL loss, but with a reversed cumulative sum applied to the output of the neural network.

### C.2.3 Ranked Probability Scoring (RPS)

Instead of focusing only on an element's event time, RPS loss instead applies to the entire time horizon, defined as:

$$
\begin{aligned}
\mathcal{L}_{RPS} = \sum_{i=1}^{B} \delta_i \sum_{k=1}^{T} (\hat{S}(k|X_i) - 1_{k<t_i})^2 \\
+ (1 - \delta_i) \sum_{k=1}^{t_i} (\hat{S}(k|X_i) - 1)^2
\end{aligned}
\tag{13}
$$

For uncensored individuals, the survival probability is pushed toward 1 at all times before the event time (rather than just *at* the event time) and is averaged over the entire time horizon. Then, for censored individuals, RPS loss still moves the survival probability towards 1 at all times before censorship, but then is only averaged up to the time of censorship $(t_i)$ since the outcome is unknown after such time.

### C.2.4 Ranking loss methods

Ranking loss is analogous to concordance index, in that it penalizes the model for an incorrect ordering of two concordant individuals. Formally, it is defined as:

$$
\mathcal{L}_{ranking} = \sum_{i \neq j}^{B} A_{i,j} \exp \frac{-(\hat{F}(t_i|X_i) - \hat{F}(t_i|X_j))}{\zeta}
\tag{14}
$$

Some works (Lee et al., 2018; Kamran & Wiens, 2021b) have demonstrated that the inclusion on such loss can increase the discriminative performance of survival models.

### C.2.5 Representation learning methods

There has been recent interest in applying contrastive learning-based losses to the survival context to improve the model's overall understanding of the training data, and therefore improve the survival performance. To evaluate the OOD performance of such models, we include SurvRNC (Saeed et al., 2024) and TE-SSL (Thrasher et al., 2024) in our baselines.

Since contrastive loss methods require a second, augmented view of the input batch, we apply the following to our data: For MRIs, we use random flip with a probability of 50%, and random affine to randomly rotate the input by 90 degrees in each direction. For the Surv* experiments, we apply color jitter and random crop. Finally, we use SCARF feature corruption (Bahri et al., 2022) for the nonimaging datasets, with a corruption rate of 60%. SurvRNC and TE-SSL are then applied regularization functions with DeepHit loss as the prognosis function. Specifically $\mathcal{L}_{DeepHit} + \gamma_1 \mathcal{L}_{SurvRNC}$ and $\mathcal{L}_{DeepHit} + \gamma_2 \mathcal{L}_{TESSL}$, respectfully. We define the $\gamma$ values in Table A2.

### C.2.6 A note on the performance of SurvRNC and TE-SSL

We observe in our experiments that SurvRNC and TE-SSL both underperformed compared to the rest of the baselines. We believe that this could indicate that contrastive learning based survival methods may generally be sensitive to hyperparameter and environment configurations.

| Layer | Details |
|---|---|
| Linear | (num features, 32) |
| ReLU | - |
| BatchNorm1D | - |
| Dropout | 10% |
| Linear | (32, 32) |
| ReLU | - |
| BatchNorm1D | - |
| Dropout | 10% |
| Linear | (32, 10) |

Table A3: Details for non-imaging MLP network.

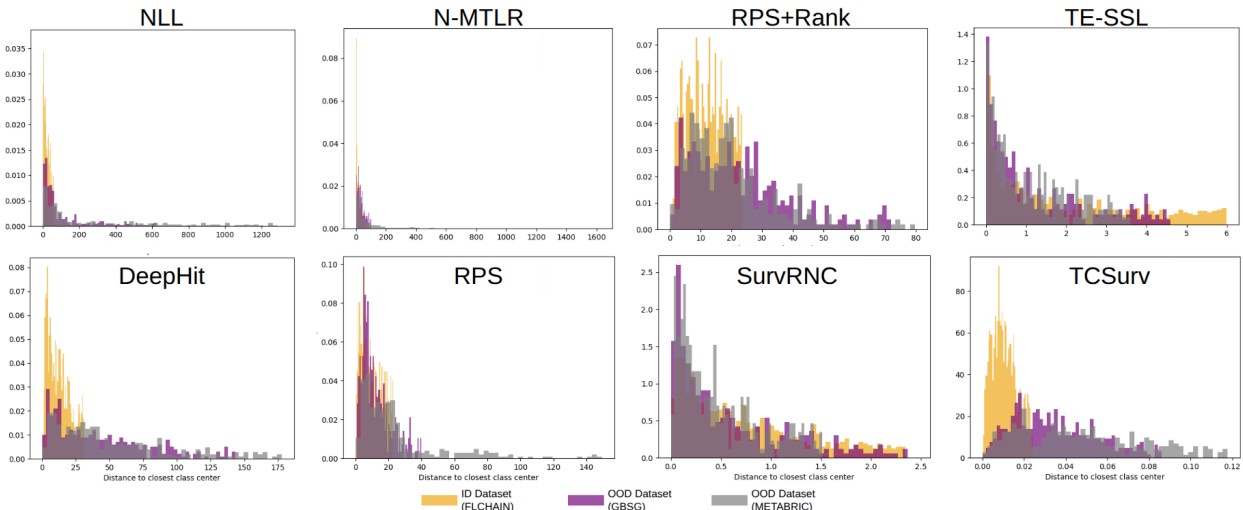

Figure A2: Histograms showing the distance of elements to their closest center for FLCHAIN experiment.

### C.3    Architecture details

#### C.3.1    3D CNN for ADNI data

The 3D CNN used for the ADNI data takes $X \in \mathbb{R}^{B \times 96 \times 96 \times 96}$ and uses a series of four convolutional layers to create the embedding $z \in \mathbb{R}^{B \times 1024}$, where $B$ is the batch size. This network utilizes kernel sizes that are unconventionally small compared to other models such as ResNet to preserve fine-grained details in the MRI data, which has been demonstrated to improve AD classification performance. Refer to (Liu et al., 2020) for further details.

#### C.3.2    MLP for non-imaging data

We use a simple multi-layer perceptron (MLP) network from (Kvamme & Ørnulf Borgan, 2019) for the non-imaging data. Table A3 provides complete details.

## D    Additional Results

We present the additional distance histograms for experiments with FLCHAIN as the ID dataset in Figure A2. Here we can clearly see that TCSurv creates the tightest clusters around the time anchors and embeds OOD samples further away, thereby improving overall OOD detection. The baselines, on the other hand, tend to create a more spread representation space, making OOD detection more challenging.

