# OpenReview forum: "TCSurv: Time-based Clustering for Reliable Survival Analysis"
_TMLR — Rejected by TMLR_

### Review · Reviewer_tyMB · 2026-02-07

**Summary Of Contributions:**

## Summary
The paper studies out-of-distribution detection under the context of survival analysis. The authors proposed TCSurv, a framework utilizing embedding-based clustering methods to identify representative clusters in the training data and is claimed to be able to filter out-of-distribution outliers.

## Strength
From a pure methodological sense, handling OOD detection problem on survival data, which are potentially stopped point processes, is an interesting problem

## Weaknesses
### Methodological concerns
It appears to me that OOD detection in survival analysis is not a very well posed problem, especially when the final decision grounds to solid findings. As we all know survival data are notoriously hard to collect, having a subgroup that deviates from the majority is a common scenario in medical studies [1], and such kind of subgroups are usually NOT discarded as being simply classified as out-of-distribution. While the authors stated in the introduction section of the paper that sometimes measurement errors such as distorted or corrupted MRI images may cause reliability issues, such kind of errors could be fixed by standard computer vision techniques, and I do NOT think they should be consider as OOD in the sense of survival analysis. Therefore, I strongly question the motivation of this paper.

### On the TCSurv framework
The proposed TCSurv framework seems to have made mostly **fundamental errors** instead of contributions. Specifically:
- In equation (4) and (5) of the paper, the authors seemed to amortize the unobserved censoring times to the (discrete) observation period and an extra censored endpoint. This approach is by no means appropriate, as modeling censoring times requires strong statistical assumptions. Moreover, sometimes the event time is even allowed to be infinite as in the case of cure models [2] which is a very important topic in modern survival analysis.
- Equation (7) emerges in an extremely weird fashion, does it even represent a formal survival function?

[1]. Rothwell, Peter M. "Subgroup analysis in randomised controlled trials: importance, indications, and interpretation." The Lancet 365.9454 (2005): 176-186.
[2]. Amico, Mailis, and Ingrid Van Keilegom. "Cure models in survival analysis." Annual Review of Statistics and Its Application 5.1 (2018): 311-342.

**Additional Comments:**

N/A

**Audience:**

No

**Audience Explanation:**

The paper is based on a problematic premise and present mostly unreadable and statistically incorrect methodologies.

**Claims And Evidence:**

No

**Claims Explanation:**

As detailed in the ``Summary Of Contributions`` section, the topic of this paper is questionable from a methodological viewpoint.

**Requested Changes:**

I recommend a strong reject to this paper, and I do not think any researchers in survival analysis would have interest in knowing the findings of this paper. No changes needed.

---

> ### Author Response · Authors · 2026-03-02
> **Response to Reviewer tyMB (1/2)**
>
> **(1) Regarding the motivation of OOD detection for survival analysis:**
> We thank the reviewer for the thoughtful considerations regarding what should and should not be considered OOD. We would like to reiterate that the goal of OOD detection is not intended to discard any information whatsoever. It is simply a mechanism which can be attached to a predictive model that provides insight as to whether the model’s prediction can be trusted, allowing for an expert end-user (e.g. clinician) to reconsider the trust they put in the model for a given input.
>
> We would also like to point out that the referenced article [1] **_precisely outlines the need for OOD detection_**, particularly in the medical domain. Deep learning models, across all domains (not just survival), frequently exhibit overfitting toward the in-distribution data and fail to provide reliable predictions of data which deviate from this distribution. Heterogeneity across subgroups with respect to risk, pathophysiology, and comorbidity [1] are all potential sources for OOD, particularly if such cases are not adequately represented within the training data.
>
> Moreover, while it is true that computer vision techniques can resolve measurement errors and/or acquisition failures to an extent, these approaches are not 100% effective at aligning corrupted data to the intended distribution [3]. As such, it is still imperative that an OOD mechanism be included, particularly for end-to-end pipelines which rely on automated reparations of potential corruptions [4]. As such, we hold our position that OOD detection is an essential component for ensuring the reliability of deep survival models with regard to image corruptions.
>
> **(2) Statistical validity of TCSurv:**
> We followed standard conventions for nonparametric deep survival analysis when designing the framework for TCSurv, which are outlined in detail in **Appendix C**.
>
> **Discrete time bins:** Nonparametric survival models are a desirable alternative to their (semi)parametric counterparts because they can predict individualized survival distributions directly, rather than relying on distributional or proportional hazards assumptions. To do this, these methods bin the time horizon into discrete intervals and then directly predict the event probability at each interval. While this does sacrifice some amount of fine-grained temporal modeling, it allows the model to capture individualized representations more effectively than other approaches. Since TCSurv is designed under the nonparametric framework, we followed this convention. We have further clarified this in our revised manuscript.
>
> **Censoring mechanism:** Nonparametric survival models can all be decomposed into two components which handle observed and censored elements respectively. Regardless of framework, all nonparametric models operate under the same general principles. For observed elements, logits are maximized at the interval which directly corresponds to event observation. Meanwhile for censored elements, all reported methods maximize the sum of all logits after the time of censorship. We deliberately modeled TCSurv after these conventions to ensure consistency with the literature in nonparametric deep survival analysis. However, we do concede that the censoring mechanism for TCSurv is more restrictive than the unbounded approaches utilized by the baseline methods (discussed in Section 5.2.1). Regardless, the reported results clearly demonstrate that TCSurv is able to achieve competitive survival performance with the baseline approaches despite these limitations. We hold that TCSurv remains grounded in standard conventions for nonparametric deep survival analysis.
>
> **(3) Infinite event times in Cure Models:**
> We also thank the reviewer for the interesting discussion surrounding cure models. While cure rates (and subsequent infinite event times) are important considerations in developing a robust survival model, we note that these methods require dramatically different design considerations and appear to be an emerging topic within deep learning. To the best of our knowledge, there does not currently exist any work which tackles the unique challenges involved in cure models with respect to nonparametric deep survival analysis (the baseline models included in our study would also _NOT_ satisfy these criteria for a cure model). **This suggests that cure modeling is a very important direction for future work in general and not a fundamental flaw with TCSurv.**

---

> > ### Author Response · Authors · 2026-03-02
> > **Response to Reviewer tyMB (2/2)**
> >
> > **(4) Formulation of the survival function**
> > Thank you for pointing this out. Firstly, we have identified a typo in our expression for **Eq (7)** (now **Eq. (8)** after manuscript edits) and have addressed the issue. We clarify that the coded implementation of the survival function is correct, meaning that this error is inconsequential with respect to the experimental outcomes, but the proper form is as follows:
> >
> > $S(t|X) = 1 - \\sum_{i=0}^t \\sigma(d)_i$
> >
> > Regardless, the survival function is derived directly from standard survival relationships [5]. However, we do acknowledge that this was not properly discussed in the manuscript and have revised **Sections 3.1** and **3.4.1** to provide more information on the formulation of the survival function.
> >
> > Under the nonparametric survival framework, a model predicts a probability mass function (PMF), which provides a likelihood of event occurrence at each discrete point in time:
> >
> > $PMF = \\phi(f_{\\theta}(X))$
> >
> > Where $\\phi(\\cdot)$ is the Softmax function. The Cumulative Incidence Function (CIF) can then be readily computed as the cumulative sum of the predicted PMF, expressed as
> >
> > $F(t|X) = \\sum_{i=0}^t \\phi(f_\\theta(X))$
> >
> > Finally, the predicted individualized survival distribution is simply the inverse of $F(t|X)$, or $S(t|X) = 1 - F(t|X)$ [5]. Since TCSurv is optimized to place embeddings close to corresponding anchor points, the PMF can be instead computed as the softmin of the distance vector.
> >
> > $PMF_{TCSurv} = \\sigma(d)$
> >
> > Where $\\sigma(\\cdot)$ is the Softmin function. Softmin results in higher probabilities for lower values. So, since the distance vector d will be minimized at the predicted event time, Softmin(d) will be maximized at such time. This results in a PMF which can conveniently be transformed into a survival function via the properties detailed above, specifically,
> >
> > $F(t|X) = \\sum_{i=0}^t \\sigma(d)$
> >
> > $S(t|X) = 1 - F(t|X)$
> >
> >
> >
> >
> >
> >
> > [1]. Rothwell, Peter M. "Subgroup analysis in randomised controlled trials: importance, indications, and interpretation." The Lancet 365.9454 (2005): 176-186.
> >
> > [2]. Amico, Mailis, and Ingrid Van Keilegom. "Cure models in survival analysis." Annual Review of Statistics and Its Application 5.1 (2018): 311-342.
> >
> > [3] Antun, Vegard et al. “On instabilities of deep learning in image reconstruction and the potential costs of AI.” Proceedings of the National Academy of Sciences (2020): 30088-30095
> >
> > [4] Huijben, Evi. et al. “Enhancing Reconstruction-Based Out-of-Distribution Detection in Brain MRI with Model and Metric Ensembles.”  Computer Methods and Programs in Biomedicine (2025)
> >
> > [5] Harvard Kvamme and Ornulf Borgan. “Continuous and discrete-time survival prediction with neural networks.” Lifetime Data Analysis (2021). 710-736

---

### Review · Reviewer_wdLm · 2026-02-13

**Summary Of Contributions:**

*Disclaimer*: I am not an expert when it comes to (deep) survival analysis. While I am quite confident with OOD detection, and in particular uncertainty-based OOD detection, it might be very well that I missed related work etc. when it comes to the survival analysis part.

The paper proposes a deep survival analysis framework that casts time-to-event prediction as a time-conditioned clustering problem in a learned embedding space, with time-bin anchors that serve as prototypes for event times and an explicit mechanism to incorporate censoring by encouraging censored samples to align with plausible future anchors. The method derives both a survival prediction (via distance-based soft assignment to time anchors) and an OOD detection score based on whether a test sample lies close to any anchor, and it evaluates this approach across imaging and tabular survival datasets under several dataset/modality/corruption shifts. The authors report improved OOD detection while maintaining competitive survival performance.

*Strengths:*

Let me begin by highlighting some of the (core) strengths of the paper. First of all, the paper is very well written and quite easy to follow (even if, as in my case, one is not familiar with the details of deep survival analysis). All tables, and visualizations (e.g. in the Appendices) are nicely formatted and make sense.

In my opinion one of the most convincing part of the paper is that the main mechanism (learning time-specific clusters around prototypes) is conceptually simple and easy to connect to survival time bins, which helps interpretability compared to more opaque survival heads.

Further, the loss explicitly accounts for censoring by enforcing compatibility with future time anchors and separation from past anchors, which is a principled way to respect the partial ordering induced by censoring (not the only way, though).

One problem that I am usually having in the uncertainty literature (where OOD tasks are part of an evaluation protocol to check how good the proposed method is) the methods (for detecting OOD in particular) are not necessarily justified why they SHOULD be appropriate for this task.  In this paper, the OOD score is linked to whether an embedding coherently fits the anchor structure, aligning the detector with the model’s inductive bias (one might debate whether later assumptions about what constitutes an OOD assumption is something suitable to assume, but at least the score itself seems to be well justified).

While the paper has no theoretical contributions (in the sense of formal properties etc.), it seems to be quite strong empirically, since experiments cover imaging and non-imaging survival data and multiple OOD constructions (modality shift, synthetic corruption, cross-dataset shift), which strengthens the narrative and the favorable performance. I like it a lot.

*Weaknesses:*

Section 3.3 mixes $0$-based and $1$-based indexing (time bins vs anchors), uses $j=0$ while anchors are defined from $1$, and the censored-loss normalizations appear inconsistent when summing to $T+1$. These issues make the method harder to reproduce and may hide edge-case bugs.

Coming back to the earlier mentioned OOD assumption, distance-to-prototype OOD detection can fail for near-OOD shifts or when the network maps OOD inputs into confident regions near anchors. The paper adopts this assumption largely heuristically (validated empirically but without strong justification or failure-mode analysis).

Relocating anchors using correctly predicted samples adds a non–end-to-end step that can affect both survival prediction and OOD scoring. Extending the same procedure to baselines for fairness is reasonable in spirit but depends on model-specific definitions what we mean by correct, which may bias comparisons.

Lastly, the mapping from softmin distances to $\hat S(t\mid X)$ is presented briefly and appears unconventional. The paper would benefit from clearer justification, guarantees (range/monotonicity), and explanation of how discretization choices impact calibration and discrimination.

**Audience:**

Yes

**Audience Explanation:**

This paper targets a practical and timely question, which is relevant to TMLR readers working on reliable ML for healthcare and other time-to-event application, so yes.

**Broader Impact Concerns:**

No broader impact concerns.

**Claims And Evidence:**

Yes

**Claims Explanation:**

Yes, but I would say with caveats.

The main claims are backed by the experimental results. However, the evidence is less clear than it could be because of the OOD assumption (OOD points being farther from anchors) is justified primarily empirically, with limited analysis of sensitivity and failure modes. A part from this I believe, as indicated, that the claims are supported by what is presented.

**Requested Changes:**

As highlighted earlier, fix and clarify indexing/notation in Section 3.3 (time bins vs anchors). Also correct the normalization/counting in the censored losses $L_{AC}$ and $L_{TC}$.

Currently OOD is primarily defined by experimental constructions. A short taxonomy and which types your method aims to detect would  improve positioning. Also an other suggestion would be to include qualitative or quantitative analysis of cases where OOD might still map close to anchors (near-OOD, etc.), and when the detector could fail. This would make the OOD argument more convincing.

Compare $S_c(x)$ against simpler scores like $\min_k d_k$ and energy-style scores based on $-\log\sum_k e^{-d_k}$. This would clarify whether the additional factor $(1-\sigma(d)_k)$ is essential.

One might want to revisit (or at least mention) the fairness of baseline adaptation (construct anchors for baselines). The idea is reasonable, but it depends on model-specific notions of what means correctly predicted.

Some typos:

I guess it should be "time-based" in the abstract.

In 5. Discussion, last sentence of the p.10 "uur" > "our".

---

> ### Author Response · Authors · 2026-03-02
> **Response to Reviewer wdLm (1/2)**
>
> We appreciate that the reviewer found our work to be both well written and empirically strong from an OOD detection perspective. In the following, we systematically respond to all major, then minor concerns raised by the reviewer.
>
> **(1) On the end-to-end nature of TCSurv:**
> Our anchor relocation strategy is agnostic to OOD data, meaning it can be performed entirely using the ID data immediately after training. In practice, one could consider assigning anchor positions using either the validation set, or a held-out “calibration” set. Once relocated, the anchor positions can be saved as a frozen model parameter and does not need to be updated. For this reason, we believe that this provides a convenient end-to-end pipeline for training, calibrating, and deploying TCSurv.
>
> **(2) Formulation of the survival function:**
> We agree that the formulation of the survival function was not well justified in the manuscript and have updated **Sections 3.1** and **3.4.1** to provide more details on this matter. We have also corrected a typo in the formulation of **Eq. (8)**, but assure the reviewer that, despite this written error, all experiments were conducted using the intended function.
>
> In nonparametric deep survival analysis, it is conventional to obtain the predicted probability mass function by taking the softmax of the model’s output ($softmax( f_\\theta(X)$ ) ). This results in a vector which contains the probability of event occurrence at each point in time. The cumulative hazard function $F(t|X)$ is simply the cumulative sum of these softmax values, which provides the probability of event occurrence at or before some time t. Finally, the survival function is found to be the inverse of the cumulative hazard ( $S(t|X) = 1 - F(t|X)$ ), which provides the probability of survival at or before some time t. For TCSurv, event probabilities are based on the proximity to different anchor points. If the model places an embedding close to an anchor $a^*_t$, then the PMF should be maximized at that point. We thus obtain the PMF using the softmin of the distance vector, which results in probabilities being maximized when the embedding is close to an anchor and minimized when it is far. From here, the cumulative hazard and survival functions can be found identically to standard convention, as expressed in **Eq. (8)**.
>
> **(3) Fairness of baseline adaption:**
> In general, all implemented survival models seek to learn a probability mass function which specifies the probability of event occurrence at some point in time. Thus, in a similar fashion to TCSurv, we can easily consider a prediction to be correct if the probability is maximized at the event time (for observed elements) or any time after censorship (for censored elements). We have revised **Section 3.4.2** to include this important clarification.
>
> **(4) OOD Taxonomy:**
> Thank you for this suggestion. We have included a brief description of OOD sources in **Section 2.2**.
>
> **(5) Minor concerns:**
> We thank the reviewer for identifying certain typos and notation misalignments. We have updated **Section 3.3** to ensure that all equations are 0-indexed and corrected the other identified typos.

---

> > ### Author Response · Authors · 2026-03-02
> > **Response to Reviewer wdLm (2/2)**
> >
> > **(6) min(d) and energy-style scoring mechanisms:**
> > We first argue that energy-style detection methods may not be well suited for survival analysis,  given the unique modeling requirements associated with censored data. Specifically, energy-based scorers rely on the idea that cross-entropy objectives in classification ultimately push the logit energy at the corresponding class label down, while all other logits see an increase in energy during training. However, survival objectives do not necessarily follow this same principle. Rather, censored data are often optimized as the sum of all logits after the time of censorship, meaning that there is no single logit that is properly minimized with respect to energy. Thus, while energy principles can be applied on observed data (being that they maximize a single logit), censored elements introduce complexity to the modeling process that deviates from what is expected by an energy-based scorer.
> >
> > Nonetheless, we include the results from additional experiments using energy and min(d) based scoring mechanisms, and found that for the **min(d)** detector, most models performed poorly across all experiments, with a few notable exceptions. Firstly, we found that N-MTLR performed well on the ADNI-T2 detection task, however subsequent performance across Bias and both nonimaging experiments suggest that overall, min(d) is not a stable detector for N-MTLR. Additionally, DeepHit initially appears to be a strong performer, particularly for the Bias analysis, but we note a very high variance, indicating that this result is not stable. Otherwise, for the nonimaging experiments, TCSurv remains the stronger performer.
> >
> > As for the energy detector, we found that, again, DeepHit initially appears strong for the ADNI-T1 suite of experiments. But it has a large variance, indicating instability in its performance. In fact, energy scoring fails across all experiments, reinforcing our claim that it is not a suitable detector for the survival setting.
> >
> >
> > |               | ADNI                  |                       | FLCHAIN               |                       |
> > |---------------|-----------------------|-----------------------|-----------------------|-----------------------|
> > |               | ADNI-T2               | Bias                  | GBSG                  | METABRIC              |
> > | NLL           | 0.2650 +/- 0.0483     | 0.4375 +/- 0.0127     | 0.6699 +/- 0.0407     | 0.7025 +/- 0.0526     |
> > | N-MTLR        | **0.7516 +/- 0.0584** | 0.5180 +/- 0.0191     | 0.6654 +/- 0.0356     | 0.6149 +/- 0.0479     |
> > | RPS           | 0.5861 +/- 0.0814     | 0.4743 +/- 0.0412     | 0.5671 +/- 0.0343     | 0.6179 +/- 0.0170     |
> > | RPS + Ranking | 0.4604 +/- 0.0715     | 0.5026 +/- 0.0123     | 0.5813 +/- 0.0506     | 0.5855 +/- 0.0608     |
> > | DeepHit       | 0.5093 +/- 0.1674     | **0.6101 +/- 0.1486** | 0.5600 +/- 0.0394     | 0.5951 +/- 0.0685     |
> > | SurvRNC       | 0.6247 +/- 0.0843     | 0.4252 +/- 0.1137     | 0.4661 +/- 0.0129     | 0.4583 +/- 0.0451     |
> > | TESSL         | 0.5671 +/- 0.1174     | 0.3244 +/- 0.0992     | 0.4454 +/- 0.0161     | 0.3822 +/- 0.0262     |
> > | TCSurv        | 0.6952 +/- 0.0999     | 0.4619 +/- 0.0356     | **0.7532 +/- 0.0344** | **0.8027 +/- 0.0129** |
> >
> > *Table 1: Analysis of min(d) detector for ADNI and nonimaging data*
> >
> > |               | ADNI                  |                       | FLCHAIN               |                       |
> > |---------------|-----------------------|-----------------------|-----------------------|-----------------------|
> > |               | ADNI-T2               | Bias                  | GBSG                  | METABRIC              |
> > | NLL           | 0.4867 +/- 0.0979     | 0.4551 +/- 0.0429     | 0.4708 +/- 0.0360     | 0.4517 +/- 0.1053     |
> > | N-MTLR        | 0.2906 +/- 0.0915     | 0.4268 +/- 0.0433     | 0.3841 +/- 0.0660     | 0.4034 +/- 0.0668     |
> > | RPS           | 0.3354 +/- 0.1021     | 0.5107 +/- 0.0725     | 0.4607 +/- 0.0417     | 0.4446 +/- 0.0526     |
> > | RPS + Ranking | 0.3783 +/- 0.0444     | 0.4486 +/- 0.0394     | **0.4885 +/- 0.0274** | 0.4729 +/- 0.0796     |
> > | DeepHit       | **0.7295 +/- 0.2175** | **0.6325 +/- 0.1797** | 0.4836 +/- 0.0256     | 0.3965 +/- 0.0598     |
> > | SurvRNC       | 0.2651 +/- 0.0854     | 0.4003 +/- 0.0914     | 0.4833 +/- 0.0021     | **0.4843 +/- 0.0097** |
> > | TESSL         | 0.4536 +/- 0.0986     | 0.4478 +/- 0.0910     | 0.4740 +/- 0.0169     | 0.4674 +/- 0.0333     |
> > | TCSurv        | 0.6210 +/- 0.1948     | 0.4430 +/- 0.0225     | 0.4781 +/- 0.0338     | 0.4503 +/- 0.0564     |
> >
> > *Table 2: Analysis of Energy detector for ADNI and nonimaging data.*

---

### Review · Reviewer_5NZ6 · 2026-02-16

**Summary Of Contributions:**

This paper presents a method for out-of-distribution (OOD) detection in the field of survival analysis in healthcare. The paper argues that this is the first paper to address this topic. They present Time-based Clustering for OOD detection in survival analysis (TCSurv). TCSurv initializes cluster centers for in-distribution (ID) data, and censored data are grouped in a cluster beyond their censorship time. This creates time-based clusters that anchor model predictions for both observed (ID) and censored data (OOD).



Strengths:
- It models both the ID and OOD separately in terms of loss calculation. Thus, the model can focus on OOD data, achieving good performance on OOD.
- Although the model performs well on OOD, it does not compromise on ID performance, maintaining comparative results with the baselines.
- They show their performance in a sufficient number of datasets.

Weakness
- The paper mentions in the introduction that there is a unique challenge in survival analysis pertaining to temporal dynamics and censored data when compared to static datasets. It would benefit the reader if the author could expand a line or two to explain the exact challenge, rather than being very general and just stating the temporal aspect.
- The paper argues that this is the first work to explicitly address OOD in survival analysis. However, I found previous literature [1, 2] that focuses on this exact topic. I found these by simply searching for keywords on Google Scholar. I assume there would be more such works, which raises concerns about the claim to be the first.
- The paper argues that OOD detection leads to reliability. Is that the only measure for reliability? I assume there are other facets of reliability that may not be as important as OOD detection but warrant discussion in the paper to make it a self-contained read, such as metrics that are valid under censoring, uncertainty quantification, etc. A paragraph in the related works or appendix would help the reader immensely to understand the reliability concept in survival analysis.
- Figure 1 is never called in the main text. The paper should use Figure 1 to explain its method in the main text.
- What is the motivation behind proposing equations 1, 2, 3, 4, and 5? This is not clear. How did they come into existence? Moreover, why $\lambda$ is used with $L_A$ and not $L_T$? How did the authors define $L_T$ and $L_A$? Why $L_{TC}$ and $L_{AC}$ have the specified form?
- Why is $\lambda_1$ = $\lambda_2$?
- In Figure 2, the caption does not explain the figure at all. Which are the ID samples and which are the corresponding OOD samples?
- In section 3.4.2, it is mentioned that “prior works in the classification domain Miller et al. (2021) have considered the combination of d and softmin(·)function to achieve a more robust score function. Hence, we also consider a similar strategy for designing the score function for evaluating OOD detection with TCSurv as” However, the paper mentions in the introduction that survival analysis is highly challenging compared to the classification task in OOD. Then how does this score function justify for survival analysis? Also, discuss briefly how the specified function is robust.
- The dataset used in this study is not highly skewed, with the maximum skewness being 69.9% for the FLCHAIN dataset. From my understanding, in the practical application of healthcare, the skewness would be high. Are there real datasets with high skewness, and how would TCSurv perform there?
- The baselines considered are old methods. The most recent included method is from 2024. Please include new methods like [1, 2]. Please also look for other recent methods for comparison.
- What is the reason for choosing the different specified backbone models for different datasets?
- In Figure 3, the range of the x- and y-axis is different among the images. Thus, the visual comparison is neither evident nor fair. Make the ranges the same. What does the y-axis denote in this figure?
- Are the results statistically significant? In section 4.2, it is mentioned that “on ADNI, a real-world dataset for Alzheimer’s dementia, TCSurv achieves the best performance”. This may not be true, since the performance of TCSurv (0.8798 ± 0.17) may not be statistically significant compared to (0.8668 ± 0.025). Similarly, perform statistically significant tests for all the results.
- In Figure 5, with increasing censorship ratio, what is the insight for the AUROC of TCSurv not being monotonic?

Minor comments:
- Please correct the citation style. Many citations of this paper should be within brackets, not inline. The readability of this paper, due to inline citations, was very poor.
- What does censor data mean? It is first defined in section 3. I urge you to include at least one line explaining censored data in the introduction.
- The concordance index is mentioned in Section 3.4.1 without any explanation. It is then explained in the Appendix. So, please refer to the appropriate appendix in section 3.4.1.
- Suggestion to keep section 4.1 as a Dataset with 4.0.1, 4.0.2, and 4.0.3 as its subsections.
- Typo: In section 4.0.3, “by setting” is repeated twice.
- Typo: In section 5, “uur”.


References:

[1] Tosaki, T., Uchino, E., Kojima, R. et al. Out-of-distribution reject option method for dataset shift problem in early disease onset prediction. Sci Rep 15, 19240 (2025). https://doi.org/10.1038/s41598-025-01811-8

[2] Zhang, X., Mehta, D., Hu, Y. et al. Adaptive transformer modelling of density function for nonparametric survival analysis. Mach Learn 114, 31 (2025). https://doi.org/10.1007/s10994-024-06686-w

**Audience:**

Yes

**Audience Explanation:**

1. The paper shows that time-based clustering may help in detecting OOD samples better than the existing methods. I believe this may be interesting to the TMLR's audience.

**Broader Impact Concerns:**

No concern

**Claims And Evidence:**

No

**Claims Explanation:**

1. The paper states that they are the first to work on OOD in survival analysis. However, there are recent methods [1,2] that work on a similar topic.
2. The recent baseline methods are not included. The most recent comparison method dates back to 2024.
3. Without a statistically significant test, it is not appropriate to claim superiority of the method in terms of performance.


References:

[1] Tosaki, T., Uchino, E., Kojima, R. et al. Out-of-distribution reject option method for dataset shift problem in early disease onset prediction. Sci Rep 15, 19240 (2025). https://doi.org/10.1038/s41598-025-01811-8

[2] Zhang, X., Mehta, D., Hu, Y. et al. Adaptive transformer modelling of density function for nonparametric survival analysis. Mach Learn 114, 31 (2025). https://doi.org/10.1007/s10994-024-06686-w

**Requested Changes:**

Weakness
- The paper mentions in the introduction that there is a unique challenge in survival analysis pertaining to temporal dynamics and censored data when compared to static datasets. It would benefit the reader if the author could expand a line or two to explain the exact challenge, rather than being very general and just stating the temporal aspect.
- The paper argues that this is the first work to explicitly address OOD in survival analysis. However, I found previous literature [1, 2] that focuses on this exact topic. I found these by simply searching for keywords on Google Scholar. I assume there would be more such works, which raises concerns about the claim to be the first.
- The paper argues that OOD detection leads to reliability. Is that the only measure for reliability? I assume there are other facets of reliability that may not be as important as OOD detection but warrant discussion in the paper to make it a self-contained read, such as metrics that are valid under censoring, uncertainty quantification, etc. A paragraph in the related works or appendix would help the reader immensely to understand the reliability concept in survival analysis.
- Figure 1 is never called in the main text. The paper should use Figure 1 to explain its method in the main text.
- What is the motivation behind proposing equations 1, 2, 3, 4, and 5? This is not clear. How did they come into existence? Moreover, why $\lambda$ is used with $L_A$ and not $L_T$? How did the authors define $L_T$ and $L_A$? Why $L_{TC}$ and $L_{AC}$ have the specified form?
- Why is $\lambda_1$ = $\lambda_2$?
- In Figure 2, the caption does not explain the figure at all. Which are the ID samples and which are the corresponding OOD samples?
- In section 3.4.2, it is mentioned that “prior works in the classification domain Miller et al. (2021) have considered the combination of d and softmin(·)function to achieve a more robust score function. Hence, we also consider a similar strategy for designing the score function for evaluating OOD detection with TCSurv as” However, the paper mentions in the introduction that survival analysis is highly challenging compared to the classification task in OOD. Then how does this score function justify for survival analysis? Also, discuss briefly how the specified function is robust.
- The dataset used in this study is not highly skewed, with the maximum skewness being 69.9% for the FLCHAIN dataset. From my understanding, in the practical application of healthcare, the skewness would be high. Are there real datasets with high skewness, and how would TCSurv perform there?
- The baselines considered are old methods. The most recent included method is from 2024. Please include new methods like [1, 2]. Please also look for other recent methods for comparison.
- What is the reason for choosing the different specified backbone models for different datasets?
- In Figure 3, the range of the x- and y-axis is different among the images. Thus, the visual comparison is neither evident nor fair. Make the ranges the same. What does the y-axis denote in this figure?
- Are the results statistically significant? In section 4.2, it is mentioned that “on ADNI, a real-world dataset for Alzheimer’s dementia, TCSurv achieves the best performance”. This may not be true, since the performance of TCSurv (0.8798 ± 0.17) may not be statistically significant compared to (0.8668 ± 0.025). Similarly, perform statistically significant tests for all the results.
- In Figure 5, with increasing censorship ratio, what is the insight for the AUROC of TCSurv not being monotonic?

---

> ### Author Response · Authors · 2026-03-02
> **Response to Reviewer 5NZ6 (1/3)**
>
> We thank the reviewer for the thorough and interesting comments. We appreciate that the reviewer found that TCSurv has strong OOD detection capabilities without compromising survival performance and that they were satisfied with the number of datasets used for our analysis. Below, we systematically address concerns raised by the reviewer.
>
> **(1) Uniqueness of OOD detection for survival analysis:**
> The requirements for constructing an objective function for nonparametric survival analysis involve aggregating predictions over multiple logits to account for censorship. This results in deviations in learning dynamics compared to those of static tasks such as classification since there is often no singular logit that corresponds directly to a prediction. Instead, the entire probability mass function contains vital survival information.
>
> We can further motivate this by considering that, unlike classification, in a survival paradigm, a uniform PMF (or survival function) can be a valid prediction, complicating the OOD detection process as many detectors rely on logit maximization at a single point.
>
> **(2) On the claim that TCSurv is the first to tackle OOD detection in survival analysis:**
> We thank the reviewer for bringing the referenced articles ([1], [2]) to our attention. However, it would appear that article [1] does not apply OOD detection within the context of survival analysis. Instead, it seems that it is more focused on classification tasks. Additionally, while article [2] proposes a survival model, it does not appear to conduct any explicit analysis on OOD detection. Thus, we believe that TCSurv is still the first approach to directly tackle OOD detection for the survival task.
>
> **(3) Inspiration for scoring mechanism from classification:**
> The challenge in OOD detection for survival analysis is that logit based detectors in classification don’t directly translate to the survival task because of the difference in prediction dynamics (as described in (1) above). Thus, the primary contribution of TCSurv is to deliberately form a representation space where the scoring function from Miller et al can be conveniently applied.
>
> As for its justification, the formulation of $d(1 - softmin(d))$ results in a minimized rejection score when embeddings have low absolute distance to an anchor point and as well as a high softmin score. Based on our analysis in **Section 5.2.1** (**Table 5**), TCSurv performs best when the value of $\\alpha$ is small. That is, the resulting hypersphere after anchor initialization is confined. Thus, minimum distance alone is less reliable since an embedding can more easily have a small distance to multiple anchors simultaneously. The additional $(1 - softmin(d))$ component mitigates this problem by rewarding embeddings which are far from all other anchors.
>
> **(4) Other forms of reliability in survival**
> We agree that including a discussion on related work in reliability for survival analysis would improve the overall comprehension and self-containment of the article. As such we have included a discussion on uncertainty quantification and explainability methods in the related works section (**Section 2.1**) to highlight the different facets of reliability in survival analysis.

---

> > ### Author Response · Authors · 2026-03-02
> > **Response to Reviewer 5NZ6 (2/3)**
> >
> > **(5) Highly skewed dataset**
> > The reviewer’s claim that real-world survival data is often highly skewed is true. To evaluate the performance of TCSurv in these domains, we have included additional experiments utilizing data from the National Alzheimer's Coordinating Center (NACC) which has a censorship rate of 76.8%. NACC is a nonimaging dataset that contains 98 unique features, making it substantially higher complexity than the benchmark nonimaging data. Since we could not utilize FLCHAIN, GBSG, or METABRIC as an OOD scenario for the NACC experiment without severe padding, we simulate an OOD setting by randomly permuting the feature columns. Full details of NACC and the OOD setting can be now found in **Appendix B.2.4**.
> >
> > Similar to the previously included experiments, TCSurv maintains strong, competitive survival performance relative to the baselines while achieving superior OOD detection scores.
> >
> > |               | C-td (Survival performance) | AUROC (OOD performance) |
> > |---------------|-----------------------------|-------------------------|
> > | NLL           | **0.8737 +/- 0.0011**       | 0.4440 +/- 0.0473       |
> > | N-MTLR        | 0.8721 +/- 0.0004           | 0.3961 +/- 0.1346       |
> > | RPS           | 0.8696 +/- 0.0005           | 0.5870 +/- 0.0598       |
> > | RPS + Ranking | 0.8694 +/- 0.0010           | 0.5253 +/- 0.0967       |
> > | DeepHit       | 0.8720 +/- 0.0003           | 0.3412 +/- 0.0755       |
> > | SurvRNC       | 0.8605 +/- 0.0002           | 0.6078 +/- 0.0840       |
> > | TESSL         | 0.8508 +/- 0.0135           | 0.5201 +/- 0.0297       |
> > | TCSurv        | 0.8684 +/- 0.0025           | **0.7226 +/- 0.0772**   |
> >
> > _Table 1: Analysis of survival and OOD performance for highly censored data (NACC)_
> >
> >
> > **(6) Old Baselines:**
> > Much of the recent development in survival analysis techniques are often applied, incremental, or designed for parametric/semi-parametric frameworks, which are not suitable for the nonparametric formulation of TCSurv. We have identified these baselines as the most state-of-the-art standalone survival techniques, which have been widely used in many application-based tasks. As for the referenced articles, article [1] appears to be more focused on classification rather than survival analysis, making it unsuitable for comparison against survival models. Additionally, the method proposed in article [2] seems to have been published in September 2024, whereas SurvRNC and TE-SSL, included in our study, were both published later, in October 2024.
> >
> > **(7) Visual comparison of distance histograms**
> > It is true that the axes are not aligned across the plots showcased in **Figure 3**. However, the purpose of this figure is to analyze the general behavior of each baseline. Namely, we are interested in identifying whether each individual model creates robust separation between ID and OOD elements. Variation in scale with respect to the x-axis is expected given the nuances in learning objective between models, but we argue that this does **not** impact the overall takeaway from the figure.
> >
> > **(8) Statistical significance of ADNI results:**
> > Thank you for bringing this up. After double checking the claim with a Wilcoxon one-sided test, we found that the performance improvement by TCSurv is _not statistically significant_ in this case. We have revised our claims in the updated manuscript.

---

> > > ### Author Response · Authors · 2026-03-02
> > > **Response to Reviewer 5NZ6 (3/3)**
> > >
> > > **(9) Insights regarding non-monotonic nature of Figure 5:**
> > > We were surprised by the non-monotonic nature of the AUROC in Figure 5 as well. We believe that this has to do with our anchor relocation scheme, particularly for censored elements. After training, the anchor points are relocated to the mean location for all correctly predicted elements. In the case of observed elements, this is a very rigid and well-defined problem: a prediction is correct if the embedding is closest to the anchor which corresponds to the observed event time. For censored elements, the formulation is more nebulous: a prediction is correct if the embedding is close to any anchor after the time of censorship. This means that for low-censorship environments, anchor relocation is actually somewhat more challenging as there are more one-to-one mappings required to correctly predict for the larger number of observed elements.
> > >
> > > As censorship increases, there are more opportunities for censored embeddings to be considered “correctly predicted”, allowing anchors to be pulled closer to the ID clusters. For this reason, there is a slight increase in performance under moderate levels of censorship.
> > > Then, as censorship further increases, the training objective itself leads to a representation space that is not as well formed in the first place, resulting in lower overall OOD detection performance.
> > >
> > > We also believe this behavior is more apparent for the OrganMNIST suite of experiments since the OOD signals are much closer to the training distribution (being that they are the same semantic object) whereas the PathMNIST suite of experiments is easier, resulting in a generally more robust representation space independent of censorship.
> > >
> > > **(10) Equations 1-5:**
> > > We were motivated by the core hypothesis that OOD detection requires explicit shaping of the representation space such that elements are embedded near temporal anchor locations. $\\mathcal{L}\_T$ (**Eq. 3**) enforces separation between anchors, which helps ensure adequate separation between ID and (potential) OOD data. Meanwhile, $\\mathcal{L}\_A$ (**Eq. 4**) pulls the embedding closer to the anchor which corresponds to the event time, supporting tight clusters.
> > > $\\mathcal{L}\_{TC}$ and $\\mathcal{L}_\{AC}$ essentially take the average of the loss across all subsequent anchors for censored components, effectively applying **Eqs. 3** and **4** over the horizon after censorship.
> > >
> > > Temporal survival data is inherently very dynamic in nature. Thus, we believe that placing too much emphasis on tight clusters (enforced by the $\\mathcal{L}_A$ components) would limit TCSurv’s ability to properly model the transitory nature of progression data. This argument is supported by the ablation of $\\lambda$ Table 5, where it can be seen that larger values of $\\lambda$ tend to improve OOD detection performance, while simultaneously reducing C-td. We have included this discussion in **Sections 3.3** and **5.2.1** for clarity.
> > >
> > > Finally, while $\\lambda_1$ and $\\lambda_2$ can be tuned individually to place more emphasis on anchor loss for observed or censored elements, we believe that in most cases, they can simply be made equal. We did not find much difference when we vary them, but it could still be a hyperparameter that could be explored in certain situations.
> > >
> > > **(11) Other comments:**
> > > Thank you for pointing out that **Figure 1** was not referenced in the paper, the caption for **Figure 2** is uninformative, the error in citation style, and some other miscellaneous omissions and typos. We have addressed all of these points in the updated manuscript. We also appreciate the suggestion to create a defined “Dataset” section and have included that in the updated draft as well.
> > >
> > > [1] Tosaki, T., Uchino, E., Kojima, R. et al. Out-of-distribution reject option method for dataset shift problem in early disease onset prediction. Sci Rep 15, 19240 (2025). https://doi.org/10.1038/s41598-025-01811-8
> > >
> > > [2] Zhang, X., Mehta, D., Hu, Y. et al. Adaptive transformer modelling of density function for nonparametric survival analysis. Mach Learn 114, 31 (2025). https://doi.org/10.1007/s10994-024-06686-w

---

### Decision · Action_Editor_Cd5j · 2026-03-26

**Recommendation:** Reject

**Additional Comments:**

Overall the paper could be resubmitted to TMLR at a later time with major revisions.

Following the already started revision in terms of more thorough comparison and mathematical clarification, it is suggested to extend the relation to survival analysis, justify use cases and provide deeper links to the survival analysis community beyond empirical numbers.

**Audience:**

Yes

**Audience Explanation:**

Yes, the topic itself could be of interest to the readership of TMLR. At the same time, it is at the interface of deep learning and survival analysis, where reviewers have diverging opinions on whether the latter community would find this work of potential interest. As such, the readership is not fully clear, as the machine learning advances themselves are less substantial in terms of fundamental OOD detection than the application context.

**Claims And Evidence:**

No

**Claims Explanation:**

In general, many parts of the paper are well supported, convincing and clear. However, at the same time, some portions were pointed out by reviewers to require improvements. For instance, the claims on novelty need to be better substantiated, the use-cases and scenario should be clarified, and methodological descriptions be made more precise. One of the reviewers remains at a recommendation of strong rejection on the account of methodological gaps in terms of links to common survival analysis methods and its literature, even after the rebuttal has been posted. Another reviewer remarks that the paper largely relies on empirical assessment to substantiate its claim and would benefit from more foundational contributions.

**Resubmission Of Major Revision:**

The authors may consider submitting a major revision at a later time.